# Triples as the Key: Structuring Makes Decomposition and Verification Easier in LLM-based TableQA

**Zhen Yang**[1,2,3], **Ziwei Du**[1,2,3], **Minghan Zhang**[1,2,3], **Wei Du**[1,2,3],
**Jie Chen**[1,2,3], **Zhen Duan**[1,2,3], **Shu Zhao**[1,2,3]*

[1]School of Computer Science and Technology, Anhui University
[2]Information Materials and Intelligent Sensing Laboratory of Anhui Province, Anhui University
[3]Anhui Provincial Key Laboratory of Security Artificial Intelligence, Anhui University
*`zhaoshuzs2002@hotmail.com`

## Abstract

As the mainstream approach, LLMs have been widely applied and researched in TableQA tasks. Currently, the core of LLM-based TableQA methods typically include three phases: question decomposition, sub-question TableQA reasoning, and answer verification. However, several challenges remain in this process: i) Sub-questions generated by these methods often exhibit significant gaps with the original question due to critical information overlooked during the LLM's direct decomposition; ii) Verification of answers is typically challenging because LLMs tend to generate optimal responses during self-correct. To address these challenges, we propose a **T**riple-**I**nspired **D**ecomposition and v**E**rification **(TIDE)** strategy, which leverages the structural properties of triples to assist in decomposition and verification in TableQA. The inherent structure of triples (head entity, relation, tail entity) requires the LLM to extract as many entities and relations from the question as possible. Unlike direct decomposition methods that may overlook key information, our transformed sub-questions using triples encompass more critical details. Additionally, this explicit structure facilitates verification. By comparing the triples derived from the answers with those from the question decomposition, we can achieve easier and more straightforward verification than when relying on the LLM's self-correct tendencies. By employing triples alongside established LLM modes, Direct Prompting and Agent modes, TIDE achieves state-of-the-art performance across multiple TableQA datasets, demonstrating the effectiveness of our method. We release our code here.

## 1 Introduction

Tabular data is ubiquitous in daily life and plays a crucial role in fields such as finance and education. Table Question Answering (TableQA) assists in judgment, analysis, and decision-making based on this data (Wang et al., 2023; Wu et al., 2024; Cao et al., 2023; Jin et al., 2023; Lee et al., 2023). TableQA involves comprehending and analyzing table content to infer answers to specific questions. With the development of large language models (LLMs) (Zhao et al., 2023b; Yang et al., 2024; Xiao et al., 2024; Lu et al., 2024; Brown et al., 2020), the use of LLMs for TableQA has become a mainstream research direction (Zhang et al., 2024b; Ye et al., 2024; Zhao et al., 2023a). Most LLM-based TableQA methods can be divided into three phases: question decomposition (Lewkowycz et al., 2022; Liu et al., 2022; Fu et al., 2022; Pasupat & Liang, 2015), sub-question TableQA reasoning and answer verification (Madaan et al., 2024; Ni et al., 2023).

---

*Corresponding authors

In recent years, under decomposition-reasoning-verification phases, LLM-based TableQA tasks have mainly employed two modes for solving complex questions: Direct Prompting (DP), a textual reasoning mode, and Agent, a symbolic reasoning mode. In the DP mode, the decomposition-reasoning-verification process primarily utilizes a few examples to prompt LLMs. Specifically, during the decomposition phase, the LLM is prompted to break down the question using manually crafted Chain-of-Thought (CoT) examples (Zhang et al., 2023b; Guan et al., 2024; Sarkar & Lausen, 2023), SQL statements (Mouravieff et al., 2024), or even zero-shot requests to leverage its capabilities directly (Zhang et al., 2023a; Gemmell & Dalton, 2023; Ye et al., 2023a; Kojima et al., 2022). In the reasoning phase, the LLM is prompted to reason through the sub-questions, while the verification phase is guided by manually constructed verification examples or by asking the LLM to re-answer for confirmation. In the Agent mode, the decomposition-reasoning-verification process is carried out automatically by the LLM (Zhang et al., 2024a; Li et al., 2024a; Gong et al., 2020; Li et al., 2024b). Specifically, during the decomposition phase, the model first observes the current and previous states to plan the next steps, designing the question incrementally. In the reasoning phase, it employs symbolic code, such as Python or SQL, to take action on the sub-questions. Finally, in the verification phase, the model performs self-correct, prompting the LLM to reassess its results and make necessary corrections.

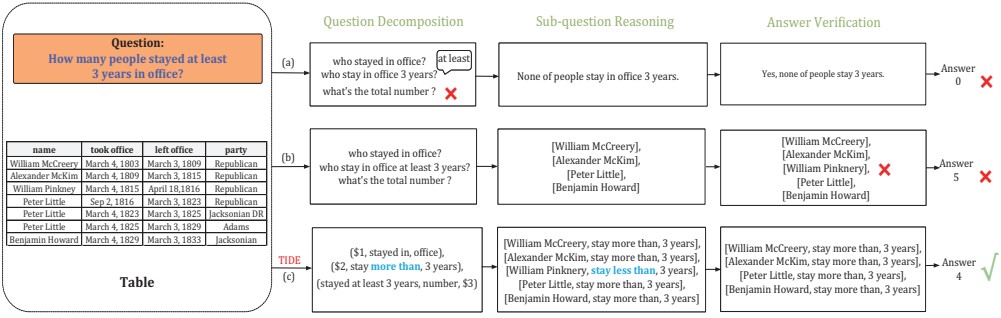

Figure 1: (a) Direct Decomposition Methods: Direct decomposition by large models can overlook critical information, resulting in a semantic gap between sub-questions and the original question. (b) Self-correct Methods:The LLM's self-correct tends to generate responses that align with the prompt, contradict the actual context, making it difficult to verification answers correctly due to this bias. (c) Our TIDE: The structured nature of triples prompts the LLM to extract as many entities and relations from the question as possible, thereby covering more critical information. Additionally, the triple structure enables a clear verification process by comparing the triples from question decomposition with those derived from the answer.

However, current LLM-based methods encounter several challenges in the decomposition-reasoning-verification process. First, the limited understanding of complex queries by LLMs often leads to fragmented sub-questions during direct decomposition, creating a significant semantic gap with the original question due to overlooked critical information. As illustrated in Figure 1(a), both Direct Prompting and Agent's direct decomposition methods frequently miss essential details, exacerbating this gap. Second, during the verification phase, simply prompting the LLM to self-correct or bypass verification hinders the accurate identification of errors in intermediate answers. Research (Huang et al., 2024) indicates that this approach may bias the model away from generating optimal responses to the initial prompt, resulting in a performance decline. As shown in Figure 1(b), direct verification of answers by LLMs often fails to detect such errors. Therefore, a more explicit method is needed to enable LLMs to generate as many sub-questions as necessary to comprehensively address the initial query while mitigating the tendency for self-correct.

To address the aforementioned challenges, we propose a **T**riple-**I**nspired **D**ecomposition and v**E**rification **(TIDE)** method that leverages the structured properties of triples for decomposition and verification, as illustrated in Figure 1(c). First, LLMs directly decomposing sub-questions often overlook critical information. In contrast, we use triples, with their fixed structure of (head entity, relation, tail entity), compels LLMs to extract as many entities and relations from the question as possible, thereby covering more key information. Second, LLMs' self-correction tends to generate responses related to the prompt, which can introduce biases in answer verification. By converting answers into triples and comparing their structures with those of the decomposed question triples,

we utilize the explicit nature of triples to determine the correctness of the answers by checking the equivalence of entities and relations. Since our triples are applied during both the decomposition and verification, and are compatible with both DP and Agent decomposition-reasoning-verification modes, our method can be utilized independently or in a joint approach with either DP or Agent.

We evaluated our method on the WikiTableQuestions (Pasupat & Liang, 2015) and TabFact (Chen et al., 2020) datasets. Our method can be applied independently to either DP or Agent methods, or in a joint reasoning framework, to enhance accuracy. Integrating our TIDE into these modes led to significant accuracy improvements, achieving the latest state-of-the-art (SOTA) results. The main contributions of this study are summarized as follows:

- We leverage the structured characteristics of triples (head entity, relation, tail entity) to prompt the LLM to extract as many entities and relations from the question as possible, thereby covering more key information. Consequently, the sub-questions derived from these triples help minimize the semantic gap with the original question.

- Utilizing the clear structure of triples allows for more easier and straightforward answer verification. By converting the answer into triple format and comparing the entities and relations with those from the question decomposition, we can directly validate the answer, thereby avoiding the biases inherent in LLMs when directly verifying responses.

- The use of triples for decomposition and verification aligns well with both DP and Agent modes, allowing our TIDE approach to be employed in either standalone or joint reasoning. Through joint reasoning, TIDE achieves state-of-the-art results in TableQA, demonstrating its effectiveness and accuracy.

## 2 RELATED WORK

### 2.1 TABLEQA

As a crucial task in table-related research, TableQA involves understanding table to infer answers and has gained significant attention in the NLP (Cheng et al., 2024; Li et al., 2023; Chen et al., 2024). With the development of LLMs, leveraging their knowledge and capabilities for TableQA become the mainstream approach (Pal et al., 2023; Lee et al., 2024; Zhong et al., 2017; Xia et al., 2023). LLM-based TableQA can be divided into three phases: question decomposition, sub-question TableQA reasoning, and answer verification. First, LLMs decompose complex questions, then reason for the sub-questions, and finally verify the intermediate answers. Under this decomposition-reasoning-verification framework, LLM-based TableQA tasks primarily employ two modes: Direct Prompting (DP) (Sui et al., 2023; Chen, 2023) and Agent modes (Li et al., 2024b; Lei et al., 2023).

### 2.2 DIRECT PROMPTING IN TABLEQA

In the DP mode, the decomposition-reasoning-verification process heavily relies on manually constructed Chain-of-Thought (CoT) prompts (Kong et al., 2024; Zhao et al., 2023d; Deng et al., 2024) to guide LLMs in gradual decomposition, reasoning, and verification. CoT was originally utilized in LLM-based text inference, incorporating prompts like think step by step to facilitate gradual reasoning toward the final answer. During the critical question decomposition phase, early DP (Zhao et al., 2023c; Sui et al., 2023; Chemmengath et al., 2021) employed few-shot examples, SQL statements, or zero-shot prompts to assist LLMs in breaking down complex questions. For instance, (Luo et al., 2023) used manually constructed CoTs and retrieval-based reconstruction to simulate human-like reasoning. BINDER (Cheng et al., 2023) guided models to reason through decomposed sub-queries using SQL statements with embedded sub-queries. Similarly, DATER (Ye et al., 2023b) employed SQL to parse, fill, and reasoning answers after extracting relevant rows and columns from large tables. (Liu et al., 2024a) added the zero-shot prompt think step by step for question decomposition and inference. In answer verification phase, the DP mode frequently neglects the verification process or asks the LLM to re-check answers directly. Studies have shown (Stechly et al., 2023; Valmeekam et al., 2023) that it is challenging for LLMs to identify issues in answers they generated themselves.

Although these LLM-based TableQA methods utilize LLMs to generate intermediate sub-questions, relying solely on a few examples for direct decomposition usually result in fragmented sub-questions

that overlook critical information. This leads to a significant semantic gap between the decomposed sub-questions and the original question. Additionally, due to LLM's tendency, the constructed verification examples prompt the LLM to direct self-correct, leading to a bias toward generating contradictory viewpoints, which hampers the accurate verification of answers in TableQA.

## 2.3 AGENT IN TABLEQA

With the continuous development of agents, their application in TableQA has been widely studied (Li et al., 2024a; Gong et al., 2020). Agents automate the decomposition-reasoning-verification process using LLMs. Typical LLM-based Agent approaches either iteratively generate intermediate tables during question decomposition or utilize LLMs to automatically break down questions. CHAIN-OF-TABLE (Wang et al., 2024) decomposes to generate intermediate tables and introduces functions to derive the answer. Similarly, ReAcTable (Zhang et al., 2024c) continuously generates intermediate tables to answer questions. (Liu et al., 2024b) used SQL statements to construct answers from newly created tables, while (Mouravieff et al., 2024) converted intermediate tables into computational graphs for reasoning. During the sub-question TableQA reasoning phase, agents typically generate code using languages like Python to obtain answers. In the answer verification phase, they often require the LLM to observe the previous and current states to perform self-correct.

The above methods typically focus on specific parts, such as decomposition or verification. To make the decomposition-reasoning-verification process more intelligent, subsequent research has developed various automated agent frameworks. StructGPT (Jiang et al., 2023) provided a unified framework for LLMs to conduct QA on structured data by invoking functions and serializing responses. However, its effectiveness was limited by the absence of integrated symbolic reasoning. To address this, SheetCopilot (Li et al., 2024a) and DataCopilot (Zhang et al., 2024a), inspired by AutoGPT, proposed solutions that traditional programming finds difficult to achieve. Nonetheless, these frameworks still require rigorous evaluation across various scenarios. To adapt to different contexts, (Liu et al., 2024a) explored table structures and introduced an agent for flexible reasoning.

Although Agent mode can handle TableQA tasks, it primarily relies on the LLM's self-decomposition and self-correct. The LLM's limited understanding of complex queries often leads to the omission of key information, resulting in a semantic gap between sub-questions and the original question. Additionally, relying on the LLM to automatically self-correct makes it difficult to detect errors in the answers. Therefore, we need a more comprehensive decomposition approach to narrow the gap between sub-questions and the original question, while also facilitating the correct verification of sub-question answers.

To address this, we leverage the structured nature of triples to assist LLMs in TableQA decomposition and verification. By extracting as many entities and relations as possible from the question to form triples, we can capture more critical information, reducing the semantic gap. Additionally, the clear structure of triples allows for easier verification by comparing the consistency between the triples formed during question decomposition and those generated from the answers. This enables more straightforward and accurate answer verification.

## 3 TASK DEFINITION

### 3.1 TABLEQA

TableQA aims to infer the answer to a question based on table content. Given a question $Q$ and a table context $C$, the goal of TableQA is to determine the final answer $A_{final}$. This task can be formally represented by Eq.1, where $C$ is structured data composed of rows and columns. Currently, most TableQA methods utilizing LLMs follow three phases: question $Q$ decomposition into sub-questions, reasoning based on table content $C$, and verification to obtain the final answer $A_{final}$. In our approach, we erialize the table content $C$ by separating each cell with a '|', resulting in the format: $value_1|value_2|...|value_n$. Here, $Q$ is a natural language question, and $A_{final}$ can be a calculated number, a specific value from the table, or any other possible outcome, as seen in the WikiTableQuestions dataset (Pasupat & Liang, 2015). Additionally, there is a specific type of TableQA known as fact verification, where $Q$ is a natural language statement about the table's content, and $A_{final}$ indicates whether the statement is correct, with possible values of *Yes* or *No*, as shown in the TabFact dataset (Chen et al., 2020).

$$LLM(Q, C) \rightarrow A_{final} \tag{1}$$

## 3.2 TRIPLE

A triple is commonly used in graph-related tasks and typically takes the form of Eq.2. Here, the head and tail are entities, while the relation represents the connection between them. When the structure is correct and a link exists, knowing any two parts of the triple allows for the inference of the unknown part based on the graph's links. For example, if the head entity and the relation are known, the unknown tail entity can be inferred by traversing the graph.

$$i - th \; triple \; (head \; entity, relation, tail \; entity) \rightarrow (e_{i1}, r_i, e_{i2}) \tag{2}$$

In our approach, we leverage the structured nature of triples in both the question decomposition and answer verification. The fixed structure of triples encourages the LLM to extract as many entities and relations from the question as possible, thereby covering more critical information. Therefore the sub-questions derived from these triples are better aligned with the original question, reducing the semantic gap. Additionally, the structured format of triples makes answer verification more easier. By comparing the entities and relations between the triples generated from the answer and those extracted during question decomposition, verification becomes more straightforward and reliable.

## 4 METHOD

**Overview.** Our method, illustrated in Figure 2, consists of four components: TIDE-Decomposition, DP/Agent Reasoning, TIDE-Verification, and Joint Reasoning. This aligns with the decomposition-reasoning-verification process found in most approaches. We utilize triples to assist the LLM in both decomposition and verification, leading to the development of the TIDE-Decomposition and TIDE-Verification modules. In the sub-question reasoning phase, we integrate both Direct Prompting and Agent modes to get the answers of the sub-questions. Ultimately in the joint reasoning phase, we apply these two modes answers for joint reasoning.

## 4.1 QUESTION DECOMPOSITION

We leverage triples to narrow the gap between sub-questions and original question. As shown in Figure 2 TIDE-Decomposition, we first utilize the LLM to extract as many entities and relations from the question as possible to form triples. Compared to direct decomposition, these triples capture more critical information. Consequently, transforming these triples into natural language sub-questions helps to bridge the semantic gap with the original query. The process in Eq.3-4.

$$LLM(Q, C, Prompt_{triples \; generate}) \rightarrow T^Q \tag{3}$$

$$\forall i \in \{1, \ldots, k\}, T_i^Q = (e_{i1}^Q, r_i^Q, e_{i2}^Q) \in T^Q, \quad LLM(T_i^Q, \mathrm{Prompt}_{decomposition}) \rightarrow S_i \tag{4}$$

Here, $S = \{S_1, \ldots, S_i, \ldots, S_K\}$ is the set of sub-questions, $S_i$ denotes the $i$-th of the sub-questions set. $T^Q = \{T_1^Q, \ldots, T_i^Q, \ldots, T_K^Q\}$, where $T_i^Q$ is the $i$-th triple: $T_i^Q = (e_{i1}^Q, r_i^Q, e_{i2}^Q)$.

As illustrated in Figure 2, we first utilize the LLM to identify the entities and relations within the question. For the example shown, we identify the explicit entities: *people*, *office*, and *3 years*, along with the relations: *stay* and *at least 3 years*. Additionally, certain words that require prediction serve as implicit entities; for instance, *how many* can be transformed into the entity *number* for prediction, while *people* acts as the entity to be predicted. After combining all identified entities and relations into triples, we utilize the LLM to convert these triples into natural language sub-questions, facilitating further inference. For example, we form the triple ($, stay in, office), leading to Sub-question 1: "Who stayed in the office?" Next, we construct the triple ($, stay more than, 3 years) to generate Sub-question 2: "Who stayed for more than 3 years?" Finally, to align with the original question, we create the triple (number, stay at least 3 years in office, $), resulting in Sub-question 3: "What's the total number of people who stayed at least 3 years in the office?"

By using these sub-questions, we aim to cover as much critical information from the original question as possible. For each question, we utilize the LLM to extract entities and relations, construct triples, and transform them into natural language sub-questions as part of the decomposition process. This decomposition breaks down the question step-by-step, allowing the LLM to refine its reasoning process and enhance accuracy.

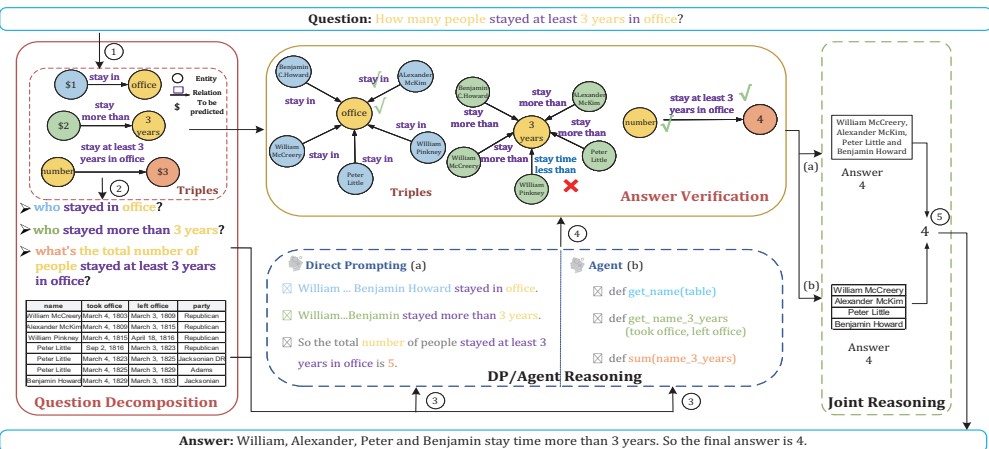

Figure 2: Question Decomposition: Our method first extracts entities and relations from the question to form triples, then divides them into sub-questions based on these triples. DP/Agent Reasoning: Next, we use either the DP or Agent mode to perform reasoning on the sub-questions. Answer Verification: In the verification phase, the inferred answers are verified based on the structure of the triples. Joint Reasoning: Finally, answers are combined using majority voting for joint reasoning.

## 4.2 DP/AGENT REASONING

Both the DP method, which utilizes CoT-related prompts, and the Agent method, which employs external tools, have proven effective for TableQA tasks. After TIDE-Decomposition, we obtain sub-questions that are processed separately through the DP and Agent approaches for inference.

In the DP mode, we prompt the LLM to infer answers for each sub-question in conjunction with the table content, resulting in intermediate answers, as shown in the Direct Prompting section of Figure 2. For the Agent method, we configure the LLM to interact with Python functions for each sub-question. Initially, it prompts the LLM to observe the current state and, using the table content, generates Python code or other symbolic languages to obtain intermediate answers, as depicted in the Agent section of Figure 2. The process can be represented as Eq.5.

$$\forall i \in \{1, \ldots, k\}, \quad LLM(S_i, \text{Prompt}_{DP}) \to A_i^{DP}, \quad LLM(S_i, \text{Prompt}_{Agent}) \to A_i^{Agent} \quad (5)$$

Here, $A^{DP} = \{A_1^{DP}, \ldots, A_i^{DP}, \ldots, A_K^{DP}\}$ is the set of DP sub-answers, $A_i^{DP}$ denotes the $i$-th of the DP sub-answers set. $A^{Agent} = \{A_1^{Agent}, \ldots, A_i^{Agent}, \ldots, A_K^{Agent}\}$ is the set of Agent sub-answers, $A_i^{Agent}$ denotes the $i$-th of the Agent sub-answers set.

## 4.3 ANSWER VERIFICATION

To mitigate the tendency of LLM self-correct towards contradictory responses, we do not directly use LLM for answer verification. Instead, we provide the triples from the question decomposition as examples, prompting the LLM to generate triples for the answers as shown in Eq.6. By comparing the relationships of the two triples and ensuring that one entity (either the head or tail entity) matches, we validate the correctness of the answer. The process can be represented as Eq.7-10.

$$\forall i \in \{1, \ldots, k\}, \quad LLM(A_i^{DP}, \text{Prompt}_{answer\ triple}) \rightarrow T_i^{DP},$$
$$LLM(A_i^{Agent}, \text{Prompt}_{answer\ triple}) \rightarrow T_i^{Agent} \qquad (6)$$

Here, $T^{DP} = \{T_1^{DP}, \ldots, T_i^{DP}, \ldots, T_K^{DP}\}$, is the set of sub-answer triples. The $i$-th DP triple $T_i^{DP} = \{(e_{i1}^{DP}, r_i^{DP}, e_{i2}^{DP})\}, i \in \{1, \ldots, k\}$. Moreover, $T^{Agent} = \{T_1^{Agent}, \ldots, T_i^{Agent}, \ldots, T_K^{Agent}\}$, is the set of sub-answer triples. The $i$-th Agent triple $T_i^{Agent} = \{(e_{i1}^{Agent}, r_i^{Agent}, e_{i2}^{Agent})\}, i \in \{1, \ldots, k\}$.

$$\forall i \in \{1, \ldots, k\}, \quad A_i^{DP} = \begin{cases} A_i^{DP} & \text{if } e_{i1}^{DP} = e_{i1}^{Q} \text{ or } e_{i2}^{DP} = e_{i2}^{Q}, r_i^{DP} = r_i^{Q}, \\ LLM(S_i, \text{Prompt}_{DP}) & \text{otherwise.} \end{cases}$$
$$(7)$$

$$\forall i \in \{1, \ldots, k\},$$
$$A_i^{Agent} = \begin{cases} A_i^{Agent} & \text{if } e_{i1}^{Agent} = e_{i1}^{Q} \text{ or } e_{i2}^{Agent} = e_{i2}^{Q}, r_i^{Agent} = r_i^{Q}, \\ LLM(S_i, \text{Prompt}_{Agent}) & \text{otherwise.} \end{cases} \qquad (8)$$

$$A_{final}^{DP} = \begin{cases} A_k^{DP} & \text{if } \forall i \in \{1, \ldots, k\}, e_{i1}^{DP} = e_{i1}^{Q} \text{ or } e_{i2}^{DP} = e_{i2}^{Q}, r_i^{DP} = r_i^{Q}, \\ LLM(S_k, \text{Prompt}_{DP}) & \text{otherwise.} \end{cases}$$
$$(9)$$

$$A_{final}^{Agent} = \begin{cases} A_k^{Agent} & \text{if } \forall\, i \in \{1, \ldots, k\}, \\ & \quad\quad e_{i1}^{Agent} = e_{i1}^{Q} \text{ or } e_{i2}^{Agent} = e_{i2}^{Q}, r_i^{Agent} = r_i^{Q}, \\ LLM(S_k, \text{Prompt}_{Agent}) & \text{otherwise.} \end{cases} \qquad (10)$$

For example, for the intermediate answer: William McCreery, Alexander McKim, William Pinkney, Peter Little, Benjamin Howard stayed in office, transforms this into the triples: (William McCreery, stay in, office), (Alexander McKim, stay in, office), (William Pinkney, stay in, office), (Peter Little, stay in, office), and (Benjamin Howard, stay in, office). Comparing with triple ($, stay in, office) which generate in question, we find both the relation and tail entity match, confirming correctness.

However, when the converted triples do not align with the initial triples, it indicates an error in the intermediate answer, prompting LLM to regenerate the answer. For instance, the transformed triple (William Pinkney, stay less than, 3 years) compared with the triple ($, stay more than, 3 years) shows the same tail entity but a different relation. Therefore, this sub-question must be re-answered. Once all intermediate answers have been verified as correct, we can derive the final answer.

### 4.4 JOINT REASONING

After reasoning and verifying both the DP and Agent modes, we obtain the final answers for each mode that pass verification. To mitigate errors from single-pass LLM reasoning, we use majority voting to combine the results. For flexibility, we first generate five answers for each mode and then randomly select a specified number based on a set hyperparameter. Further results and analysis of different selection combinations are provided in Section 5. Eq.11 $A_{final}$ is the final answer.

$$A_{final} = Majority\ Vote(A_{final}^{DP}, A_{final}^{Agent}) \qquad (11)$$

## 5 EXPERIMENT

### 5.1 DATASETS AND EVALUATION

**Dataset.** We used two widely recognized TableQA datasets: WikiTableQuestions (Pasupat & Liang, 2015) and TabFact (Chen et al., 2020). WikiTableQuestions is a commonly used dataset for complex

TableQA tasks, involving operations such as aggregation, comparison, and arithmetic calculations. Our method was evaluated on its test set, containing 4,344 table-related questions. The TabFact dataset focuses on fact verification in TableQA, determining whether a statement is correct based on the table's content, with answers being yes or no. TabFact test set includes 2,024 samples.

**Evaluation.** Following previous work (Cheng et al., 2023; Liu et al., 2024a), we use exact match accuracy as the evaluation metric, comparing whether the final answer matches the gold answer.

## 5.2 IMPLEMENTATION DETAILS

We followed previous work (Cheng et al., 2023; Ye et al., 2023b; Liu et al., 2024a) and used GPT-3.5 as the LLM, with a temperature of 0.8. To reduce errors from single-pass reasoning, we generated five answers for both the DP and Agent. The combination of answer selections is detailed in Section 5.5. The specific prompts used for the DP and Agent methods are provided in the Appendix I.

## 5.3 BASELINES

For LLM-based TableQA, we first directly compared several language models, including SASP (Ou & Liu, 2022), TAPAS-large (Eisenschlos et al., 2020), T5-3B (Xie et al., 2022), TAPEX-large (Liu et al., 2021), and Codex (Cheng et al., 2023). Additionally, to better evaluate TIDE in DP and Agent, we selected methods under both frameworks, including BINDER (Cheng et al., 2023), DATER (Ye et al., 2023b), StructGPT (Jiang et al., 2023), DTE (Wang et al., 2023), TACR (Wu et al., 2023), ITR (Lin et al., 2023), (Liu et al., 2024b), Tab-PoT (Xiao et al., 2024), CHAIN-OF-TABLE Wang et al. (2024), ReAcTable (Zhang et al., 2024c), Cabinet (Patnaik et al., 2024), and (Liu et al., 2024a). For detailed descriptions of the baseline method, please refer to Appendix H.

## 5.4 MAIN RESULTS

Table 1 presents the results on the WikiTableQuestions. Both standalone DP and agent deliver strong results. Notably, when our method combines both modes, it achieves SOTA, surpassing all other methods by 1.35%-20%. Table 2 shows the results on the TabFact. Similarly, our method is effective in both standalone DP and agent, outperforming most approaches. The combination of our method with both modes achieves SOTA performance, improving over all methods by about 1%-16%. Since best performance on TabFact is already high, the gains of 1.32% should be interpreted as a proportion of scope of further improvement possible, $1.32/(100 - 88.5)$, which is $\approx 11.48\%$.

TIDE's success largely stems from its effective guidance during decomposition and verification. In decomposition, we identify all entities and relations in the question, forming triples that ensure sub-questions align with the original semantics. This step-by-step reasoning mitigates errors from direct answering. During verification, explicit standards enable the LLM to make consistent judgments, reducing errors at each step and ensuring accurate final inferences. By seamlessly integrating DP and Agent approaches, our method leverages joint reasoning across both modes, minimizing biases from single-pass predictions.

Additionally, TIDE-Agent consistently outperforms TIDE-DP on both datasets, primarily due to two factors. First, the agent generates Python code to interact with tables, making it better suited for handling structured tabular data. Second, its global observation capability allows it to select the next action based on the current state, enabling more accurate global reasoning.

## 5.5 ABLATION STUDY

To examine TIDE's impact during the decomposition and verification phases in both DP and Agent modes, we conducted ablation experiments on the WikiTableQuestions and TabFact datasets. We evaluated the effect of using only TIDE's decomposition and only TIDE's verification in both modes, with results shown in Table 3.

As Table 3 illustrates, TIDE's decomposition has a significant impact on accuracy. This is primarily due to the pipeline's sequential nature: first, the question is decomposed, then responses are validated. If the decomposition is incorrect—leading to erroneous sub-questions or failing to capture the original question's semantics—the final answer will be incorrect, even if intermediate responses are

Table 1: Results on WikiTableQuestions. TIDE-DP and TIDE-Agent show results for DP and Agent modes separately. TIDE-DP&Agent shows the joint reasoning result.

| Method | Acc. |
|---|---|
| TAPEX-large (Liu et al., 2021) | 59.10 |
| T5-3B (Xie et al., 2022) | 50.60 |
| BINDER (Cheng et al., 2023) | 64.60 |
| DATER (Ye et al., 2023b) | 65.90 |
| StructGPT (Jiang et al., 2023) | 57.00 |
| DTE (Wang et al., 2023) | 54.20 |
| TACR (Wu et al., 2023) | 60.20 |
| ITR (Lin et al., 2023) | 63.40 |
| (Liu et al., 2024b) | 55.80 |
| CHAIN-OF-TABLE (Wang et al., 2024) | 59.94 |
| ReAcTable (Zhang et al., 2024c) | 68.00 |
| Cabinet (Patnaik et al., 2024) | 69.10 |
| (Liu et al., 2024a)-DP&Agent | 73.65 |
| TIDE-DP | 66.51 |
| TIDE-Agent | 68.72 |
| **TIDE-DP&Agent** | **75.00** |

Table 2: Results on TabFact dataset. TIDE-DP and TIDE-Agent show results for DP and Agent modes separately. TIDE-DP&Agent shows the joint reasoning result.

| Method | Acc. |
|---|---|
| TAPAS-large (Eisenschlos et al., 2020) | 81.00 |
| TAPEX-large (Liu et al., 2021) | 84.20 |
| SASP (Ou & Liu, 2022) | 77.00 |
| T5-3B (Xie et al., 2022) | 83.68 |
| Codex end-to-end (Cheng et al., 2023) | 72.60 |
| Codex SQL (Cheng et al., 2023) | 80.70 |
| BINDER (Cheng et al., 2023) | 85.10 |
| DATER (Ye et al., 2023b) | 85.60 |
| StructGPT (Jiang et al., 2023) | 87.30 |
| CHAIN-OF-TABLE (Wang et al., 2024) | 80.20 |
| ReAcTable (Zhang et al., 2024c) | 86.10 |
| Tab-PoT (Xiao et al., 2024) | 85.77 |
| (Liu et al., 2024a)-DP&Agent | 88.50 |
| TIDE-DP | 81.32 |
| TIDE-Agent | 88.19 |
| **TIDE-DP&Agent** | **89.82** |

accurate. Conversely, when decomposition is correct, the LLM's inherent reasoning ability can still maintain a certain level of accuracy in intermediate responses, even without explicit verification.

Table 3: Ablation results on WikiTableQuestions (WTQ) and TabFact.

| Method | WTQ | TabFact |
|---|---|---|
| **TIDE-DP** | **66.51** | **81.32** |
| w/o Decomposition | 60.33 (↓ 6.18) | 77.22 (↓ 4.10) |
| w/o verification | 64.89 (↓ 1.62) | 79.64 (↓ 1.68) |
| **TIDE-Agent** | **68.72** | **88.19** |
| w/o Decomposition | 61.21 (↓ 7.51) | 81.27 (↓ 6.92) |
| w/o verification | 67.12 (↓ 1.60) | 86.86 (↓ 1.33) |
| **TIDE-DP&Agent** | **75.00** | **89.82** |
| w/o Decomposition | 68.61 (↓ 6.39) | 85.47 (↓ 4.35) |
| w/o verification | 71.22 (↓ 3.78) | 87.06 (↓ 1.76) |

Table 4: Impact of answer selection in TIDE-DP and TIDE-Agent.

| Agent | DP | WTQ | TabFact |
|---|---|---|---|
| 1 | 1 | 65.06 | 81.52 |
| 3 | 3 | 73.73 | 89.33 |
| 5 | 5 | 75.00 | 89.82 |
| 1 | 3 | 67.11 | 81.99 |
| 3 | 1 | 70.28 | 83.27 |
| 1 | 5 | 69.38 | 82.16 |
| 5 | 1 | 71.04 | 87.35 |
| 3 | 5 | 73.02 | 84.98 |
| 5 | 3 | 74.15 | 89.23 |

## 5.6 PERFORMANCE ANALYSIS UNDER DIFFERENT FACTORS

**Number of Answer Selections in TIDE-DP and TIDE-Agent.** Our method can be applied independently in either DP or Agent, or used for joint reasoning across both modes. To examine the effect of different selection combinations, Table 4 presents the results. For a single mode, increasing the number of selected answers improves performance, suggesting that majority voting mitigates bias from single-pass LLM reasoning. In joint mode, performance is better when the number of Agent-generated answers exceeds DP-generated answers, indicating that the agent's results are more reliable and its reasoning capability is stronger, consistent with our analysis in Section 5.4.

**Number of Triples Decomposed in TableQA.** Generally, more triples indicate a more complex question, requiring additional inference steps. To evaluate our method's adaptability to complexity and decomposition efficiency, we report the frequency and accuracy of decomposed triples in both datasets. As shown in Figure 3, in WTQ, inference accuracy declines as question complexity increases. We speculate this is because complex questions require more evidence, straining the

LLM's ability to reason over longer texts. In contrast, TabFact exhibits fluctuating accuracy. Examining its decomposition, we found that due to limited prompt examples, TabFact questions are broken down more granularly than in WTQ. This sometimes leads to overly simple questions being split into multiple sub-questions, increasing their count. Any error in inferring a sub-question can propagate, resulting in accuracy fluctuations.

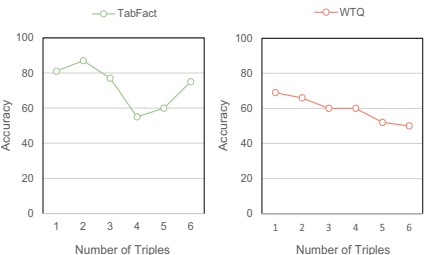 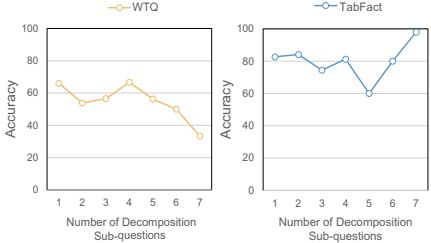

Figure 3: Number of Triples Decomposed in WTQ and TabFact.

Figure 4: Number of Sub-questions Decomposed in WTQ and TabFact.

**Number of Sub-questions Decomposed in TableQA.** We further analyze the sub-questions derived from triples. To evaluate TIDE' across both datasets, we report accuracy for different numbers of decomposed sub-questions. As shown in Figure 4, the number of sub-questions varies, with the LLM automatically breaking complex triples into additional sub-questions during inference. This is especially noticeable in the Agent. While the number of triples reflects the problem's complexity, the number of sub-questions indicates reasoning difficulty. In Figure 4, aside from fluctuations at the 4-step decomposition in WTQ, the overall trend aligns with the decline seen in Figure 3 for WTQ. Similarly, TabFact shows notable fluctuations, supporting our previous hypothesis.

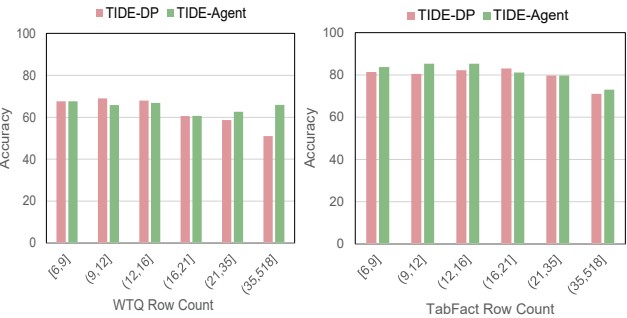

Figure 5: Impact of table size on TableQA performance.

**Table Size in TableQA.** To assess the inference capabilities of TIDE-DP and TIDE-Agent across different table sizes, we categorized tables based on previous work. Each interval contained approximately 430 data points, and we calculated average accuracy within these ranges. As shown in Figure 5, TIDE-Agent generally outperforms TIDE-DP across both datasets, consistent with our analysis in Section 5.4. With its ability to process tables via code and observe data globally, TIDE-Agent proves more effective in most cases. In WTQ, we observe that as table size increases, the performance gap between TIDE-Agent and TIDE-DP widens. Larger tables result in longer serialized text, making inference harder for TIDE-DP, whereas TIDE-Agent efficiently extracts data through code. This highlights the need for future research to focus on handling larger tables more effectively.

## 6 CONCLUSION

To address issues like overlooked information and contradictory self-corrections, we propose TIDE, which leverages the structured nature of triples. The fixed structure ensures LLMs extract more entities and relations, covering critical information. It also simplifies answer verification by directly comparing decomposed triples with those from the answer.

## 7 ACKNOWLEDGEMENTS

Our work is supported by the National Natural Science Foundation of China (62476003), Anhui Province Excellent Scientific Research and Innovation Team (2024AH010004), Anhui Provincial Natural Science Foundation - Water Science Joint Fund (2408055US006), the University Synergy Innovation Program of Anhui Province (GXXT-2023-050), and SMP-Zhipu.AI Large Model Cross-Disciplinary Fund (SMP-Zhipu20240210). We also acknowledge the support from Zhipu AI-Anhui University Joint Research Center, and the High-Performance Computing Platform of Anhui University.

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

APPENDIX

## A   ABLATED IMPLEMENT

This section mainly provides a formal description of the implementation details of the ablation experiments.

**1) w/o TIDE-Decomposition**

This method is similar to Chain of Thought (CoT)(Wei et al., 2022), where the prompt *think step by step* and few examples used. Sub-questions Decomposition $\{S_1, ..., S_K\}$ can be represented as the equation.

$$LLM(Q, C, Prompt_{CoT}) \rightarrow S_i$$

**2) w/o TIDE-Verification**

This approach resembles not providing any explicit standards. Instead, the LLM is prompted to rely on self-correct(Huang et al., 2024) using the instruction: *Review your previous answer and find problems with your answer. Based on the problems you found, improve your answer. Please reiterate your answer* and few examples. The Sub-answers $A_i$ Verification can be represented as the equation.

$$A_i = \begin{cases} A_i & \text{if } LLM(A_i, Prompt_{self}) = True, \\ LLM(S_i, Prompt_{self}) & \text{otherwise.} \end{cases}$$

## B   OTHER LLMS

**1) Open-source and Closed-source LLM Models**

Table 5 results demonstrate that our method improves the performance of multiple open-source and closed-source models, validating its effectiveness.

| Model | Acc. |
|---|---|
| Llama 2-7b | 42.23 |
| **Llama 2-7b + Ours** | **45.00 (2.77 ↑)** |
| Llama 3-70b | 53.25 |
| **Llama 3-70b + Ours** | **58.64 (5.39 ↑)** |
| GLM 4 | 60.17 |
| **GLM 4 + Ours** | **64.00 (3.83 ↑)** |
| Gemini 1.5 | 54.23 |
| **Gemini 1.5 + Ours** | **59.84 (5.61 ↑)** |
| Claude 3.5 | 70.38 |
| **Claude 3.5 + Ours** | **75.60 (5.22 ↑)** |

Table 5: Comparison of models with and without our method

| Size | Acc. |
|---|---|
| Llama 2-7b + Ours | 45.00 |
| Llama 3-8b + Ours | 50.27 |
| Llama 3-70b + Ours | 58.64 |

Table 6: Model Size and Accuracy Comparison

**2) Model Size Affect**

We conducted verification on different sizes of the open-source Llama model, and the results are as in Table 6. The results indicate that the model size has a significant impact on performance. Larger models consistently achieve better results, demonstrating improved effectiveness.

**3) Non-LLM methods**

Some baseline methods in our paper are non-LLM-based approaches. Table 7 comparison shows a substantial performance gap, with our method achieving improvements ranging from 11.7% to 30.5%. This demonstrates the effectiveness of our approach.

| Model | Acc. |
|---|---|
| Structured Attention (Wang et al., 2019) | 44.50 |
| ReasTAP-Large (Zhao et al., 2022) | 58.70 |
| TAPEX-large (Liu et al., 2021) | 59.10 |
| OmniTab-Large (Jiang et al., 2022) | 63.30 |
| Ours | 75.00 |

Table 7: Comparison of Non-LLM methods

**4) Data Contamination**

In order to mitigate the impact of LLM data contamination, we address this issue by comparing the performance of directly using the same LLM for question answering versus applying our proposed method in Table 8. From the comparison result, it is evident that our method improves the model's accuracy by approximately 27%, demonstrating that our approach itself brings significant performance gains.

| Model | Acc. |
|---|---|
| GPT Direct QA (Cheng et al., 2023) | 48.70 |
| Ours | 75.00 |

Table 8: Comparison of Direct QA.

| Method | DP | Agent | Result |
|---|---|---|---|
| (Liu et al., 2024a)-DP&Agent | 5 | 5 | 73.65 |
| **Ours** | **5** | **5** | **75.00** |

Table 9: Comparison of SOTA.

## C  VERIFICATION FAILS

This section primarily discusses cases of verification failure. In our implementation, we limit the number of iterations to a maximum of 4. Additionally, our statistics show that nearly 90% of the data reaches the final answer within a single iteration. Furthermore, less than 2% of cases fail to obtain an answer after 4 iterations, for which we set a default value of *error*. To mitigate the impact of these *error* cases, we follow the SOTA approach by obtaining multiple results from the DP and Agent modes, respectively, and applying majority voting. In Table 9, we also present the results of different answer combinations based on the two modes.

## D  DP AND AGENT JOINT

This section primarily explores the joint effect of DP and Agent. In our method, majority voting is employed for their integration. First, we compare it with methods that also utilize majority voting,

| Method | Result |
|---|---|
| ReAcTable (Zhang et al., 2024c) | 68.00 |
| **Ours** | **75.00** |

Table 10: Comparison with majority voting method.

as shown in Table 10, where it can be observed that combining DP and Agent indeed achieves excellent performance. Subsequently, we compare it with the results of using DP or Agent alone in SOTA methods, as presented in Table 11.

Our method follows the SOTA approach, which combines DP and Agent. We introduced triples , building upon the SOTA framework. As shown in the table 13, while the performance in the DP mode is roughly the same, significant improvements were observed in the Agent and joint modes. This demonstrates the effectiveness of incorporating triples.

| Method | Result |
|---|---|
| (Liu et al., 2024a)-DP [6] | 66.99 |
| **Ours-DP** | 66.51 (0.48↓ almost same) |
| (Liu et al., 2024a)-Agent [6] | 63.77 |
| **Ours-Agent** | 68.72 (4.95↑) |
| (Liu et al., 2024a)-DP&Agent [6] | 73.65 |
| **Ours-DP&Agent** | 75.00 (1.35↑) |

Table 11: Comparison with SOTA DP and Agent.

For the comparable results in the DP mode, we speculate that this is due to DP relying on text-based reasoning, which has limited understanding of the structured nature of tables. Agent, using Python-based methods, has a stronger understanding of table structure.

# E   TIME AND TOKEN USED

In practical implementation, we combine the triple prompt with the decomposition prompt and integrate answer generation with triple verification. We analyzed the average time and token usage for the decomposition and verification LLM calls in Table 12. Besides, compared to other LLM-based methods, our approach reduces the number of API calls while achieving better results, as shown in Table 13.

| Operation | Time | Token |
|---|---|---|
| Decomposition | 2.190s | Prompt length 1159 + table length + question length |
| Verification | 3.211s | Prompt length 970 + table length + question length + triple and sub-question length |

Table 12: Operation time and token details

| Methods | Result | Number of API calls |
|---|---|---|
| CHAIN-OF-TABLE (Wang et al., 2024) | 59.94 | (Next operation 1 + Argument 1 + Transform table 1) * Iter $N = 3N$ |
| Ours | 75.00 | Decompose triple and sub-questions 1 + Answer and verification 1 = 2 |

Table 13: Comparison of Methods with Results and API Calls

# F  QUESTION TYPE

Our method does not impose limitations, as it structures questions and answers into triples, making it broadly applicable to table question-answering tasks. Additionally, our triples are generated specifically based on the questions and are designed to aid in understanding the table context.

The WTQ dataset covers a variety of operations and question types, including where, which, what, who, how many, average, and is, among others.

To illustrate this, we have provided actual results from the test dataset.

---

**"Average" Type**

Question: in cycle 4 of austria's next top model, what is the average of all the contestants' ages?

Triples:  (in cycle 4 of austria's next top model, $1),
  (for each contestant, what is their age),
  ($1, is the average of, contestants' ages)

Decompose: 1. what is in cycle 4 of austria's next top model?
  2. for each contestant, what is their age?
  3. what is the average of all the contestants' ages?

---

**"Who" Type**

Question: who did the team play after the law vegas legends on november 10?

Triples: ($1, did the team play, after the law vegas legends),
  ($1, played against, $2),
  ($2, played on, november 10)

Decompose: 1. who did the team play after the law vegas legends?
  2. when did the team play against the law vegas legends?
  3. what was the date when the team played against the law vegas legends?

---

**"Which" Type**

Question: which country had the most cyclists finish within the top 10?

Triples: ($1, had, cyclists finish within, the top 10),
  (the top 10, total, cyclists),
  ($2, country, most)

Decompose: 1. which country had cyclists finish within the top 10?
  2. every country, what's the total number of cyclists who finished within the top 10?
  3. which country had the most cyclists finish within the top 10?

---

**"Was" Type**

Question: was 18th cross chronologically after moggina manasu?

Triples: (18th cross, film, $1),
  (moggina manasu, film, $2),
  (18th cross, was (chronologically), after)

Decompose: 1. what film was 18th cross?
  2. what film was moggina manasu?
  3. was 18th cross chronologically after moggina manasu?

---

Figure 6: Question type example.

# G   ERROR ANALYSIS

Here, we randomly selected 100 test samples and categorized the errors for analysis.

## G.1   ANSWER FORMATING

| Description Losses | 1939/40 | 1940/41 | 1941/42 | 1942/43 | 1943/44 | 1944/45 | Total |
|---|---|---|---|---|---|---|---|
| Direct War Losses | 360,000 | - | - | - | - | 183,000 | 543,000 |
| Murdered | 75,000 | 100,000 | 116,000 | 133,000 | 82,000 | - | 506,000 |
| Deaths In Prisons & Camps | 69,000 | 210,000 | 220,000 | 266,000 | 381,000 | - | 1,146,000 |
| Deaths Outside of Prisons & Camps | - | 42,000 | 71,000 | 142,000 | 218,000 | - | 473,000 |
| Murdered in Eastern Regions | - | - | - | - | - | 100,000 | 100,000 |
| Deaths other countries | - | - | - | - | - | - | 2,000 |
| Total | 504,000 | 352,000 | 407,000 | 541,000 | 681,000 | 270,000 | 2,770,000 |

Question: how many people were murdered in 1940/41?

Triples: ($1, were murdered in, year),
($2, number of people, $3)

Decompose: 1. who were murdered in 1940/41?
2. what's the number of people who were murdered in 1940/41?

Reasoning: The people who were murdered are described in row 1 of the table.
They were murdered in various years, including 1940/41, as indicated in the table.
The number of people murdered in 1940/41 is 100000.
Final Answer: 100000

Gold Answer: 100,000

**Error Analysis : 100000 and 100,000 format different**

Figure 7: Answer formating error.

## G.2 INCORRECT NUMERICAL COMPARISON

| Description Losses | 1939/40 | 1940/41 | 1941/42 | 1942/43 | 1943/44 | 1944/45 | Total |
|---|---|---|---|---|---|---|---|
| Direct War Losses | 360,000 | - | - | - | - | 183,000 | 543,000 |
| Murdered | 75,000 | 100,000 | 116,000 | 133,000 | 82,000 | - | 506,000 |
| Deaths In Prisons & Camps | 69,000 | 210,000 | 220,000 | 266,000 | 381,000 | - | 1,146,000 |
| Deaths Outside of Prisons & Camps | - | 42,000 | 71,000 | 142,000 | 218,000 | - | 473,000 |
| Murdered in Eastern Regions | - | - | - | - | - | 100,000 | 100,000 |
| Deaths other countries | - | - | - | - | - | - | 2,000 |
| Total | 504,000 | 352,000 | 407,000 | 541,000 | 681,000 | 270,000 | 2,770,000 |

Question: what is the only years that total losses were below 350,000?

Triples: ($1, total losses, below 350,000),
         ($2, were the only, years)

Decompose: 1. what is the total losses in each year?
           2. which years had total losses below 350,000?
           3. what is the only years that total losses were below 350,000?

Reasoning: The total losses in each year are as follows: 1939/40: 504,000; 1940/41: 352,000; 1941/42: 407,000; 1942/43: 541,000; 1943/44: 681,000; 1944/45: 270,000.
           The years that had total losses below 350,000 is: 1940/41.
           Final Answer: 1940/41

Gold Answer: 1944/45

**Error Analysis : Incorrectly assumed that 352,000 < 350,000**

Figure 8: Incorrect numerical comparison error.

### G.3 SPECIAL ROW

| Description Losses | 1939/40 | 1940/41 | 1941/42 | 1942/43 | 1943/44 | 1944/45 | Total |
|---|---|---|---|---|---|---|---|
| Direct War Losses | 360,000 | - | - | - | - | 183,000 | 543,000 |
| Murdered | 75,000 | 100,000 | 116,000 | 133,000 | 82,000 | - | 506,000 |
| Deaths In Prisons & Camps | 69,000 | 210,000 | 220,000 | 266,000 | 381,000 | - | 1,146,000 |
| Deaths Outside of Prisons & Camps | - | 42,000 | 71,000 | 142,000 | 218,000 | - | 473,000 |
| Murdered in Eastern Regions | - | - | - | - | - | 100,000 | 100,000 |
| Deaths other countries | - | - | - | - | - | - | 2,000 |
| Total | 504,000 | 352,000 | 407,000 | 541,000 | 681,000 | 270,000 | 2,770,000 |

Question: what is the last description of losses on this chart ?

Triples: ($1, had, most cyclists),
      ($2, finish within, the top 10),
      ($3, is, country)

Decompose: 1. who finished within the top 10?
        2. what's the country for each cyclist?
        3. which country had the most cyclists finish within the top 10?

Reasoning: Thought: Extract the unique values from the "Description Losses" column. Select the last
      value to answer the second sub-question.
     Action: python_repl_ast Action Input: df['Description Losses'].unique()
     Observation: ['Direct War Losses' 'Murdered' 'Deaths In Prisons & Camps' 'Deaths Outside
      of Prisons & Camps' 'Murdered in Eastern Regions' 'Deaths other countries' 'Total']
     Thought: The unique values of the "Description Losses" column are ['Direct War Losses',
      'Murdered', 'Deaths In Prisons & Camps', 'Deaths Outside of Prisons & Camps',
      'Murdered in Eastern Regions', 'Deaths other countries', 'Total']. Select the last value.
     Action: python_repl_ast Action Input: df['Description Losses'].unique()[-1]
     Observation: Total
     Final Answer: Total

Gold Answer: Deaths other countries

Error Analysis : Unable to identify special rows, such as total

Figure 9: Special row error.

## G.4 COUNTING

| Placing | Rider | Location | Country | Event | Placing | Rider | Nationality |
|---|---|---|---|---|---|---|---|
| 31 October 2008 | 2008–09 World Cup | Manchester | United Kingdom | Keirin | 2 | Jason Kenny | GBR |
| 31 October 2008 | 2008–09 World Cup | Manchester | United Kingdom | Sprint | 1 | Victoria Pendleton | GBR |
| 1 November 2008 | 2008–09 World Cup | Manchester | United Kingdom | 500 m time trial | 1 | Victoria Pendleton | GBR |
| 1 November 2008 | 2008–09 World Cup | Manchester | United Kingdom | Sprint | 1 | Jason Kenny | GBR |
| 2 November 2008 | 2008–09 World Cup | Manchester | United Kingdom | Team sprint | 1 | Jason Kenny | GBR |
| 5 November 2008 | 5th International Keirin Event | Manchester | United Kingdom | International keirin | 2 | Ross Edgar | GBR |
| 2 November 2008 | 2008–09 World Cup | Manchester | United Kingdom | Team sprint | 1 | Jamie Staff | GBR |
| 2 November 2008 | 2008–09 World Cup | Manchester | United Kingdom | Keirin | 1 | Victoria Pendleton | GBR |
| 2 November 2008 | 2008–09 World Cup | Manchester | United Kingdom | Team sprint | 1 | Ross Edgar | GBR |
| 13 February 2009 | 2008–09 World Cup | Copenhagen | Denmark | Team sprint | 1 | Chris Hoy | GBR |
| 13 February 2009 | 2008–09 World Cup | Copenhagen | Denmark | Team sprint | 1 | Jason Kenny | GBR |
| 13 February 2009 | 2008–09 World Cup | Copenhagen | Denmark | Team sprint | 1 | Jamie Staff | GBR |
| 13 February 2009 | 2008–09 World Cup | Copenhagen | Denmark | Sprint | 1 | Victoria Pendleton | GBR |
| 30 October 2009 | 2009–10 World Cup | Manchester | United Kingdom | Sprint | 1 | Victoria Pendleton | GBR |
| 30 October 2009 | 2009–10 World Cup | Manchester | United Kingdom | Sprint | 1 | Chris Hoy | GBR |
| 30 October 2009 | 2009–10 World Cup | Manchester | United Kingdom | Keirin | 1 | Chris Hoy | GBR |
| 30 October 2009 | 2009–10 World Cup | Manchester | United Kingdom | 500 m time trial | 2 | Victoria Pendleton | GBR |
| 1 November 2009 | 2009–10 World Cup | Manchester | United Kingdom | Team sprint | 1 | Chris Hoy | GBR |
| 1 November 2009 | 2009–10 World Cup | Manchester | United Kingdom | Team sprint | 1 | Ross Edgar | GBR |
| 1 November 2009 | 2009–10 World Cup | Manchester | United Kingdom | Team sprint | 1 | Jamie Staff | GBR |

Question: what is the number of 1st place finishes across all events?

Triples: ($1, is the number of, 1st place finishes),
      (across all events, total, $2)

Decompose: 1. for each event, what is the placing?
        2. what is the total number of 1st place finishes across all events?

Reasoning: In event 0, Jason Kenny placed 2nd. In event 1, Victoria Pendleton placed 1st. In event 2, Victoria Pendleton placed 1st. In event 3, Jason Kenny placed 1st. In event 4, Jason Kenny placed 1st. In event 5, Ross Edgar placed 2nd. In event 6, Jamie Staff placed 1st. In event 7, Victoria Pendleton placed 1st. In event 8, Ross Edgar placed 1st. In event 9, Chris Hoy placed 1st. In event 10, Jason Kenny placed 1st. In event 11, Jamie Staff placed 1st. In event 12, Victoria Pendleton placed 1st. In event 13, Victoria Pendleton placed 1st. In event 14, Chris Hoy placed 1st. In event 15, Chris Hoy placed 1st. In event 16, Victoria Pendleton placed 2nd. In event 17, Chris Hoy placed 1st. In event 18, Ross Edgar placed 1st. In event 19, Jamie Staff placed 1st.
The total number of 1st place finishes across all events is 14.
Final Answer: 14

Gold Answer: 17

Error Analysis : count error, total is 17

Figure 10: Counting error.

## H BASELINES

SASP (Ou & Liu, 2022) uses lexical and structural features to generate programs for solving pseudo-programs. TAPEX-large (Liu et al., 2021) learns a neural SQL executor on a synthetic corpus of executable SQL queries and their outputs. T5-3B (Xie et al., 2022) within the UnifiedSKG framework unifies 21 SKG tasks into a text-to-text format for comprehensive SKG research. TAPAS-large

(Eisenschlos et al., 2020) creates a balanced dataset of millions of automatically generated training examples for intermediate learning before fine-tuning. Codex (Cheng et al., 2023), as an OpenAI API, can generate SQL or Python statements and perform end-to-end QA.

BINDER (Cheng et al., 2023) iteratively refines pseudo-SQL queries for final answers. DATER (Ye et al., 2023b) extracts sub-tables and decomposes questions for joint reasoning. StructGPT (Jiang et al., 2023) enhances zero-shot reasoning with specialized interfaces for structured data. DTE (Wang et al., 2023) refines text-to-SQL QA using counterfactual examples. TACR (Wu et al., 2023) aligns multi-hop questions with various modalities for evidence retrieval. ITR (Lin et al., 2023) selects relevant rows and columns for compact sub-tables. (Liu et al., 2024b) creates new tables with external information for SQL-based answers. Tab-PoT (Xiao et al., 2024) enhances open-source LLMs with prompt management and post-processing modules. CHAIN-OF-TABLE Wang et al. (2024) plans operation chains dynamically based on table structure and questions. ReAcTable (Zhang et al., 2024c) iteratively generates intermediate tables using LLMs and external code. Cabinet (Patnaik et al., 2024) removes noise from tables to improve LLM reasoning accuracy. (Liu et al., 2024a) explore combining DP and Agent to address LLM sensitivity to table structure.

# I  DP AND AGENT PROMPT

## I.1  DP PROMPT

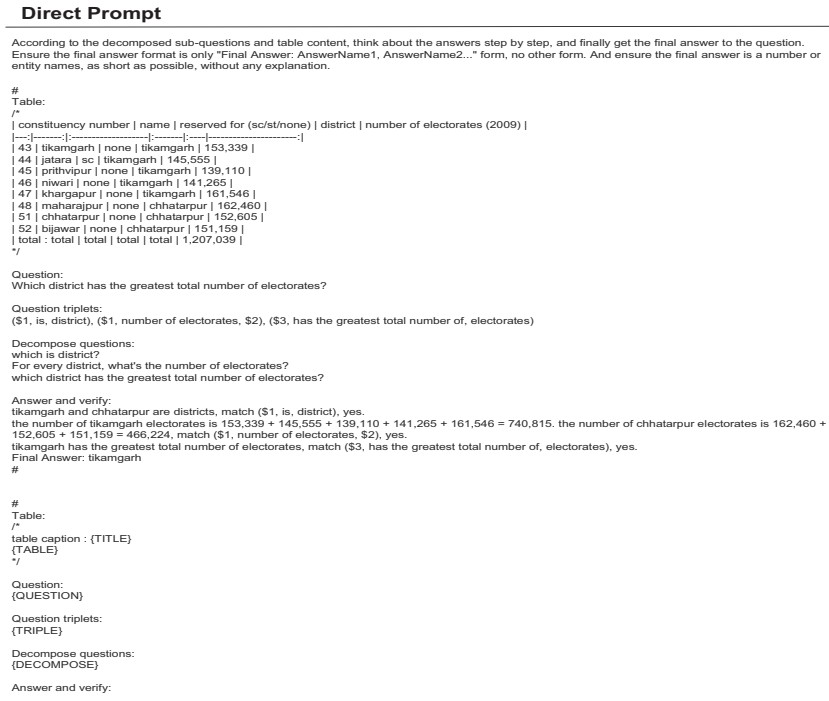

Figure 11: The prompt of Direct Prompting mode.

## I.2 AGENT PROMPT

---

**Agent Prompt**

---

You are working with a pandas dataframe in Python. The name of the dataframe is `df`. Your task is to use `python_repl_ast` to answer the question posed to you.

Tool description:
- `python_repl_ast`: A Python shell. Use this to execute python commands. Input should be a valid python command. When using this tool, sometimes the output is abbreviated - ensure it does not appear abbreviated before using it in your answer.

Guidelines:
- **Aggregated Rows**: Be cautious of rows that aggregate data such as 'total', 'sum', or 'average'. Ensure these rows do not influence your results inappropriately.
- **Data Verification**: Before concluding the final answer, always verify that your observations align with the original table and question.

Strictly follow the given format to respond:

Question: the input question you must answer
Question Triples: Triples drawn according to the question
Decompose Sub-questions: the decompose sub-questions of the question based on the question triples
Thought: you should always follow the decompose sub-questions, and always think about what to do to interact with `python_repl_ast`
Action: can **ONLY** be `python_repl_ast`
Action Input: the input code to the action
Observation: the result of the every sub-question's action whether conforms to the structure of a triple and the answer whether is right.
... (this Thought/Action/Action Input/Observation can repeat N times)
Thought: after verifying the table, observations, and the question, I am confident in the final answer
Final Answer: the final answer to the original input question (AnswerName1, AnswerName2...)

Notes for final answer:
- Ensure the final answer format is only "Final Answer: AnswerName1, AnswerName2..." form, no other form.
- Ensure the final answer is a number or entity names, as short as possible, without any explanation.
- Ensure to have a concluding thought that verifies the table, observations and the question before giving the final answer.

You are provided with a table regarding "[TITLE]". This is the result of `print(df.to_markdown())`:

[TABLE]

**Note**: All cells in the table should be considered as `object` data type, regardless of their appearance.

Begin!
Question: [QUESTION]
Question Triples: [Triples]
Decompose Sub-questions: [Decompose]
"""

---

Figure 12: The prompt of Agent mode.

