# OpenReview forum: "Triples as the Key: Structuring Makes Decomposition and Verification Easier in LLM-based TableQA"
_ICLR.cc/2025/Conference — ICLR 2025 Poster_

### Official Review · Reviewer_uGLH · 2024-10-29

**Soundness:** 2
**Presentation:** 2
**Contribution:** 2
**Rating:** 6
**Confidence:** 4

**Summary:**

The paper proposed an inference time approach, TIDE, to address table question answering. Their method builds on a Decomposition-Reasoning-Verification approach for Table QA using LLM, and they proposed to use semantic triples as intermediate representations instead of natural text. First, during question decomposition, they extract triples from the question then generate subquestions based on those triples. Then, they apply CoT (for DP mode)/generating python code (for agent mode) to answer each subquestions. Finally, in the answer verification stage, they compare the triples from stage 1 with triples from stage 2 to filter correctly answered triples.

Their contribution is using triples as intermediate representations in order to facilitates decomposition and verification, and showed that it leads to improvements over SOTA.

**Strengths:**

Incorporating triples as intermediate steps is an interesting approach, and it intuitively makes sense as a natural intermediate representation to tackle the verification of subquestions. The authors effectively combine existing ideas from different problems and apply it in the settings of table QA, showing SOTA results on challenging table qa datasets.

**Weaknesses:**

My main concern is clarity and contribution.
For clarity
1.  Details of how each module is implemented is confusing. The formulation really only has one equation (eq 1), and it is about the Table QA problem. Formal explanation (instead of examples only) for each submodule would help improve clarity. Furthermore, I think the prompt for extracting triples are not provided?

2. It’s unclear to me how the final answer is synthesized after the verification stage. Again, providing formulation for this stage would be more helpful.

3. In Table 3 of your ablation studies, you have TIDE vs TIDE-decomposition vs TIDE-verification. It is confusing to me how the ablated versions are implemented. Again, I think having formulation would help improve clarity.

In terms of contribution, since this is combining different existing ideas from other tasks (see below for examples), I would appreciate more in-depth analysis of the proposed method.

1.  Would other language models be equally effective with generating triples? When does this assumption breakdown? How does model size affect the result and error propagation through each stage ... etc. Have you tested this with open source large/small sized language models or other information extraction approaches for triple extraction? Is it possible to substitute some LLM API calls with other non-LLM methods without hindering performance?

2. The strong performance could potentially come from data contamination. How do you attribute the success of TIDE vs GPT having seen those tables in pretraining?

3. In verification stage, you mentioned the LLM is prompted to regenerate if verification failed. I would appreciate more analysis on this part such as showing how often verification fails or how many times max you limit the LLM to regenerate.


PS by combining existing ideas I mean
1. Question decomposition is not new. See [1] [2] for example
2. using LLM for information extraction is not new. See [3] for example

I think there are some merits in combining these insights and seeing that triples could apply to this problem, so I think more analysis could help improve this paper.

[1] Sewon Min, Victor Zhong, Luke Zettlemoyer, and Hannaneh Hajishirzi. Multi-hop Reading Comprehension through Question Decomposition and Rescoring. ACL 2019.

[2] Xiaoyu Yang and Xiaodan Zhu.  Exploring Decomposition for Table-based Fact Verification. EMNLP Findings 2021

[3] Zhiyuan Fan and Shizhu He. 2023. Efficient Data Learning for Open Information Extraction with Pre-trained Language Models.  EMNLP Findings 2023

**Questions:**

This goes back to the clarity of the verification stage.The example authors provided seems to be a ‘COUNT’ type questions. How does the system handle other types of numeric aggregation? For questions that are not numeric in nature (e.g. compose-style, multi-hop questions), how does the final stage work after verification? Does this approach only work on aggregation type questions? What algorithms are you using to unify the triples for each subquestion?

---

> ### Author Response · Authors · 2024-11-21
>
> Thanks for your review. Below, we provide seven detailed responses to address your concerns. We hope this will resolve your confusion and gain your approval. Your feedback has been immensely helpful, and we will update into the final version.
>
> ### 1. Details for Each Module
>
> **In the appendix of the paper, we provided examples prompts. The complete prompts can also be found in the supplementary materials.** Additionally, to provide a clearer explanation, we include the following formal description:
>
> **Task Definition**: Given a question \\(Q\\) and a table context \\(C\\), the goal is to determine the final answer $A_{final}$.
>
> $$
> LLM(Q,C) \rightarrow A_{final}
> $$
>
> **Specific Implementation Process**:
>
> 1). Generate the Question Triples:
>    The structure of each triple is (head entity, relation, tail entity), \\(i\\)-th triple formally expressed as: $(e_{i1},r_i,e_{i2})$.
> the question triple can get from:
>
> $$
> LLM(Q,C,Prompt_{triples\ generate}) \rightarrow T^{Q}
> $$
>
> Here,  $T^Q = \\{T^Q_1, \dots, T^Q_i, \dots, T^Q_K\\}$ is the question triples set, where $T^Q_i$ is the \\(i\\)-th triple:  $T^Q_i = (e^Q_{i1}, r^Q_i, e^Q_{i2})$.
>
> 2). **Decomposition into Sub-questions:**
>
> $$
> ∀ i \in \\{1, \ldots, k\\}, T^Q_i = (e^Q_{i1}, r^Q_{i}, e^Q_{i2}) \in T^Q，LLM(T^Q_i, Prompt_{decomposition}) \rightarrow S_{i}
> $$
>
> The $S=\\{S\_1, \dots, S_i, \dots, S_K\\}$ is the set of sub-questions,  $S_i$ denotes the \\(i\\)-th of the sub-questions set.
>
> 3). **Execution by DP/Agent to Obtain Sub-answers:**
> $$
> ∀ i \in \\{1, \ldots, k\\}, LLM(S_i, Prompt_{DP}) \rightarrow A_i^{DP}, LLM(S_i, Prompt_{Agent}) \rightarrow A_i^{Agent}
> $$
>
> Where $A^{DP}=\\{A_1^{DP}, \dots, A_i^{DP}, \dots, A_K^{DP}\\}$ is the set of DP sub-answers,  $A_i^{DP}$ denotes the \\(i\\)-th of the DP sub-answers set. $A^{Agent}=\\{A_1^{Agent}, \dots, A_i^{Agent}, \dots, A_K^{Agent}\\}$ is the set of Agent sub-answers,  $A_i^{Agent}$ denotes the \\(i\\)-th of the Agent sub-answers set.
>
> 4). **Sub-answers Triples Generate:**
>
> $$
> ∀ i \in \\{1, \ldots, k\\}, LLM(A_i^{DP}, Prompt_{answer\ triple}) \rightarrow T_i^{DP}, LLM(A_i^{Agent}, Prompt_{answer\ triple}) \rightarrow T_i^{Agent}
> $$
>
> Here, $T^{DP} = \\{T_{1}^{DP}, \dots, T_{i}^{DP}, \dots, T_{K}^{DP}\\}$, is the set of DP sub-answer triples. The \\(i\\)-th DP triple $T_{i}^{DP} = \\{(e_{i1}^{DP}, r_{i}^{DP}, e_{i2}^{DP})\\},  i \in \\{1, \ldots, k\\}$.
>
> Moreover, $T^{Agent} = \\{T_{1}^{Agent}, \dots, T_{i}^{Agent}, \dots, T_{K}^{Agent}\\}$, is the set of Agent sub-answer triples. The \\(i\\)-th Agent triple $T_i^{Agent} = \\{(e_{i1}^{Agent}, r_{i}^{Agent}, e_{i2}^{Agent})\\},  i \in \\{1, \ldots, k\\}$.
>
> 5). **Sub-answers Verification:**
>
> $$
> ∀ i \in \\{1, \ldots, k\\},  A_i^{DP} =
> \\begin{cases}
> A_{i}^{DP} & \text{if}\  e_{i1}^{DP} = e_{i1}^{Q}\ \text{or}\ e_{i2}^{DP} = e_{i2}^{Q}, \ r_{i}^{DP} = r_{i}^{Q}\\\
> LLM(S_i, Prompt_{DP}) & \text{otherwise}
> \\end{cases}
> $$
>
> $$
> ∀ i \in \\{1, \ldots, k\\}, A_{i}^{Agent} =
> \begin{cases}
> A_{i}^{Agent} &  \text{if}\ e_{i1}^{Agent} = e_{i1}^{Q}\ \text{or}\ e_{i2}^{Agent} = e_{i2}^{Q}, \ r_{i}^{Agent} = r_{i}^{Q} \\\
> LLM(S_i, Prompt_{Agent}) & \text{otherwise}
> \end{cases}
> $$
>
> $$
> A_{final}^{DP} =
> \begin{cases}
> A_{k}^{DP} & \text{if}\ ∀ i \in \\{1, \ldots, k\\}, e_{i1}^{DP} = e_{i1}^{Q}\  \text{or}\  e_{i2}^{DP} = e_{i2}^{Q}, r_{i}^{DP} = r_{i}^{Q} \\\
> LLM(S_k, Prompt_{DP}) & \text{otherwise}
> \end{cases}
> $$
>
> $$
> A_{final}^{Agent} =
> \begin{cases}
> A_{k}^{Agent} & \text{if } ∀i \in \\{1, \ldots, k\\}, e_{i1} ^{Agent} = e_{i1}^{Q} \text{ or } e_{i2}^{Agent} = e_{i2}^{Q}, r_{i}^{Agent} = r_{i}^{Q} \\\\
> LLM(S_k, Prompt_{Agent}) & \text{otherwise}
> \end{cases}
> $$
>
> 6). **Joint Reasoning:**
>
> $$
> A_{final} = Majority\ Vote(A_{final}^{DP},A_{final}^{Agent})
> $$
>
> Here, $A_{final}$ is the final answer of the question.
>
> ### 2. Ablated Implement
>
> ​	1). **w/o TIDE-Decomposition**: This approach is **similar to Chain of Thought (CoT)** [1], where only the prompt "think step by step" + a few examples is used.
>
> ​	[1] Wei J, Wang X, Schuurmans D, et al. Chain-of-thought prompting elicits reasoning in large language models[J]. Advances in neural information processing systems, 2022, 35: 24824-24837.
>
> The sub-questions Decomposition set without TIDE-Decomposition $S=\\{S_1, \dots, S_i, \dots, S_K\\}$ can get from:
>
> $$
> LLM(Q, C, Prompt_{CoT})\rightarrow S_i
> $$
>
> 2). **w/o TIDE-Verification**:  **The LLM is prompted to rely on self-correct** [2] using the instruction: "Review your previous answer and find problems with your answer. Please reiterate your answer" + few examples.
>
> ​	[2] Jie Huang, Xinyun Chen, Swaroop Mishra, Huaixiu Steven Zheng, Adams Wei Yu, Xinying Song, and Denny Zhou. Large language models cannot self-correct reasoning yet. In The 12th International Conference on Learning Representations, 2024.
>
> The sub-answers $A_i$ verification is:
>
> $$
> A_{i} =
> \begin{cases}
> A_{i} & \text{if } LLM(A_i, Prompt_{Agent})=True \\\\
> LLM(S_i, Prompt_{Agent}) & \text{otherwise}
> \end{cases}
> $$

---

> > ### Author Response · Authors · 2024-11-21
> >
> > ### 3. Other LLMs
> >
> > In our paper, **we only present the results of GPT-3.5 to align with prior LLM-based studies on the two datasets and ensure experimental fairness**. Here, we provide the result on other LLMs to show the efficiency of our method.
> >
> > Essentially, most text-generation LLMs can generate triples, although the quality of the generated results may vary depending on the self-performance and size of the LLM. The results are presented as follows:
> >
> > **1）Open-source and Closed-source LLM Models**
> >
> > The results demonstrate that our method improves the performance of multiple open-source and closed-source models, validating its effectiveness.
> >
> > | Model  | Acc. |
> > |-- |-- |
> > | Llama 2-7b | 42.23  |
> > | **Llama 2-7b + Ours**  | **45.00 (2.77 ↑)** |
> > | Llama 3-70b  | 53.25  |
> > | **Llama 3-70b + Ours** | **58.64 (5.39 ↑)** |
> > | GLM 4  | 60.17  |
> > | **GLM 4 + Ours**  | **64.00 (3.83 ↑)** |
> > | Gemini 1.5 | 54.23  |
> > | **Gemini 1.5 + Ours**  | **59.84 (5.61 ↑)** |
> > | Claude 3.5  | 70.38  |
> > | **Claude 3.5 + Ours**  | **75.60 (5.22 ↑)** |
> > | | |
> >
> > **2）Model Size Affect**
> >
> > We conducted verification on different sizes of the open-source Llama model, and the results are as follows. The results indicate that the model size has a significant impact on performance. **Larger models consistently achieve better results, demonstrating improved effectiveness.**
> >
> > | Size  | Acc.  |
> > | --- | -- |
> > | Llama 2-7b + ours | 45.00 |
> > | Llama 3-8b + ours | 50.27 |
> > | Llama 3-70b + ours | 58.64 |
> > | | |
> >
> > **3）Non-LLM methods**
> >
> > ​	Some baseline methods in our paper are non-LLM-based approaches. **The comparison shows a substantial performance gap, with our method achieving improvements ranging from 11.7% to 30.5%.** This demonstrates the effectiveness of our approach.
> >
> > | Model | Acc.  |
> > | ------- | ----- |
> > | Structured Attention [3] | 44.50 |
> > | ReasTAP-Large [4]  | 58.70 |
> > | TAPEX-large [5] | 59.10 |
> > | OmniTab-Large [6]  | 63.30 |
> > | Ours  | 75.00 |
> > | | |
> >
> > [3] Wang B, Titov I, Lapata M. Learning Semantic Parsers from Denotations with Latent Structured Alignments and Abstract Programs[C]//Proceedings of the 2019 Conference on Empirical Methods in Natural Language Processing and the 9th International Joint Conference on Natural Language Processing (EMNLP-IJCNLP). 2019: 3774-3785.
> >
> > [4] Zhao Y, Nan L, Qi Z, et al. ReasTAP: Injecting Table Reasoning Skills During Pre-training via Synthetic Reasoning Examples[C]//Proceedings of the 2022 Conference on Empirical Methods in Natural Language Processing. 2022: 9006-9018.
> >
> > [5] Liu Q, Chen B, Guo J, et al. TAPEX: Table Pre-training via Learning a Neural SQL Executor[C]//International Conference on Learning Representations.
> >
> > [6] Jiang Z, Mao Y, He P, et al. OmniTab: Pretraining with Natural and Synthetic Data for Few-shot Table-based Question Answering[C]//Proceedings of the 2022 Conference of the North American Chapter of the Association for Computational Linguistics: Human Language Technologies. 2022: 932-942.
> >
> > ### 4. Data Contamination
> >
> > Assuming your concern is valid, we address it by **comparing the performance of directly using the same LLM for question answering versus applying our method**. The results of direct question answering can be found in the cited paper below, and the comparison results are shown in the following table.
> >
> > From the comparison, **it is evident that our method improves the model's accuracy by approximately 27%**, demonstrating that our approach itself brings significant performance gains.
> >
> > | Model  | Acc. |
> > | ---- | :---- |
> > | GPT Direct QA [5] | 48.70 |
> > | Ours  | 75.00 |
> > | | |
> >
> > [5] Zhoujun Cheng, Tianbao Xie, Peng Shi, Chengzu Li, Rahul Nadkarni, Yushi Hu, Caiming Xiong, Dragomir Radev, Mari Ostendorf, Luke Zettlemoyer, et al. Binding language models in symbolic languages. In The 11th International Conference on Learning Representations, 2023.
> >
> > ### 5. Verification Fails
> >
> > In our implementation, we limit the number of iterations to **a maximum of 4**.
> >
> > Additionally, our statistics show that **nearly 90% of the data reaches the final answer within a single iteration**.
> >
> > Furthermore, less than 2% of cases fail to obtain an answer after 4 iterations, for which we set a default value of 'error'.
> >
> > To mitigate the impact of these 'error' cases, we follow the SOTA approach by obtaining multiple results from the DP and Agent modes, respectively, and **applying majority voting**.
> >
> > In Table 4 of our paper, we also present the results of different answer combinations based on the two modes. The comparison with the baseline SOTA results is as follows:
> >
> > | Method | DP | Agent | Result  |
> > | ---- | ---- | --- | --- |
> > | (Liu et al., 2024a)-DP&Agent [6] | 5 | 5  | 73.65  |
> > | **Ours** | **5** | **5** | **75.00** |
> >
> > ​	[6] Tianyang Liu, Fei Wang, and Muhao Chen. Rethinking tabular data understanding with large language models. In Proceedings of the 2024 Conference of the North American Chapter of the Association for Computational Linguistics: Human Language Technologies (Volume 1: Long Papers), pp. 450–482, 2024a.

---

> ### Author Response · Authors · 2024-11-21
>
> ### 6. Question types
>
> **Not limited to count-type calculation problems. Our method does not impose limitations, as it structures questions and answers into triples, making it broadly applicable to table question-answering tasks.**
>
> The WTQ dataset we tested **includes a variety of question types, such as *where*, *which*, *what*, *who*, *how many*, and *is*.** The TabFact dataset, on the other hand, focuses on determining whether a statement is true based on the table's content.
>
> To illustrate this more clearly, we provide actual results obtained from the test dataset:
>
> | question         | which country had the most cyclists finish within the top 10? |
> | ---------------- | ------------------------------------------------------------ |
> | question triples | (\\$1, had, cyclists finish within, the top 10), (the top 10, total, cyclists), (\\$2, country, most) |
> | sub-questions    | 1. which country had cyclists finish within the top 10? 2. for every country, what's the total number of cyclists who finished within the top 10? 3. which country had the most cyclists finish within the top 10?" |
> | | |
>
> | question         | who did the team play after the law vegas legends on november 10? |
> | ---------------- | ------------------------------------------------------------ |
> | question triples | (\\$1, did the team play, after the law vegas legends), (\\$1, played against, \\$2), (\\$2, played on, november 10) |
> | sub-questions    | 1. who did the team play? 2. who did the team play after the law vegas legends? 3. when did the team play against the law vegas legends? 4. what was the date when the team played against the law vegas legends?|
> | | |
>
> | question         | was 18th cross chronologically after moggina manasu?         |
> | ---------------- | ------------------------------------------------------------ |
> | question triples | (18th cross, film, \\$1), (moggina manasu, film, \\$2)，(18th cross, was (chronologically), after) |
> | sub-questions    | 1. what film was 18th cross? 2. what film was moggina manasu? 3. was 18th cross chronologically after moggina manasu? |
> | | |
>
> ### 7. Question decomposition and Using LLM for Information Extraction is not New
>
>    Question decomposition and answer verification are two commonly employed techniques in TableQA. Our innovation lies in **introducing structured triples into TableQA. By leveraging this explicit and fixed structure, the task of TableQA becomes clearer and more comprehensive. Our strategy is adaptable to and enhances performance in both common TableQA frameworks, DP and Agent.**
>
>    1. Question decomposition is crucial for addressing complex queries [7]. Our approach **utilizes entity recognition to capture more comprehensive semantic information, thereby reducing the gap between sub-questions and the original semantics**.
>
>       ​	[7] Wei J, Wang X, Schuurmans D, et al. Chain-of-thought prompting elicits reasoning in large language models[J]. Advances in neural information processing systems, 2022, 35: 24824-24837.
>
>    2. Answer verification plays a significant role in enhancing QA accuracy. However, self-verification often suffers from refutability due to the lack of explicit standards [8]. **Our method provides clear verification criteria through the fixed structure of triples.**
>
> In summary, these are our responses to the seven points you raised. We hope this clarifies your concerns. If you have any further questions, please ask to us, and we will provide timely feedback.

---

> ### Author Response · Authors · 2024-11-25
> **Rebuttal Invitation**
>
> Dear Reviewer uGLH,
>
> We have provided responses to your concerns. Your suggestions have been invaluable for improving our paper, and the rebuttal content will be incorporated into the final version. If you have any further questions, please feel free to raise them. Your recognition is very important to us. We sincerely invite you to engage in the rebuttal process. Thank you!

---

> > ### Comment · Reviewer_uGLH · 2024-11-25
> >
> > Thank you for taking the time to conduct detailed experiments as suggested. Please do include the revisions as I have yet to see it show up in the pdf linked above. Given the addition of new experimental results and analyses, I am willing to update my score.

---

> > > ### Author Response · Authors · 2024-11-28
> > >
> > > Dear Reviewer uGLH,
> > >
> > > We have updated all the rebuttal responses in the newly submitted PDF. Due to page limitations, the formalized description of the method has been included in the main method section, while other content in the appendix. Thank you again for your review and recognition; your suggestions have been extremely helpful in improving our paper. Thank you!

---

### Official Review · Reviewer_BvQy · 2024-11-03

**Soundness:** 4
**Presentation:** 3
**Contribution:** 3
**Rating:** 6
**Confidence:** 3

**Summary:**

This paper proposed a decomposition and verification strategy based on triple to address TableQA tasks. Specifically, it decomposes and transforms questions using triples to capture more key information, while utilizes explicit structure for better verification. By combining LLM-based reasoning and symbolic reasoning based on structured representations, TIDE improves the performance in the TableQA task

**Strengths:**

1. The motivation of the work is clear. It leverages triple extraction as a tool to enhance understanding and validation of TableQA questions.

2. The experiment is sufficient, which compares TIDE with many TableQA baselines. Further study can reveal the properties of the proposed method.

**Weaknesses:**

1. The contribution is limited; it only proposes a framework based on LLMs with triple extraction. It would be better if it could be demonstrated on some open-source models like Llama3.

2. The conclusion drawn from the experimental results that TIDE improves the effectiveness of TableQA problems is questionable. The results of DP and Agent baseline for both datasets are not SOTA. So, can it be considered that the gain in this result comes from additional reasoning integration of DP and Agen rather than the introduction of triplet?

**Questions:**

The appendix lacks cases, including successful cases compared to other methods, error cases analysis, etc. I think showing some cases can better explain the working mechanism of the entire pipeline.

---

> ### Author Response · Authors · 2024-11-21
>
> We sincerely appreciate your review and recognition of our work. Your suggestions have been instrumental in improving our paper, and we will incorporate additional details in the final version to enrich its content. Below are our two responses and supplementary explanations to address your feedback.
>
> ### 1. Experiments with Other LLMs
>
> In our paper, **we primarily present the results of GPT-3.5 to maintain experimental fairness and align with prior LLM-based studies on the two datasets**. Here, we provide the result on other LLMs to show the efficiency of our method.
>
> #### 1) Open-Source and Closed-Source Models
>
> The results demonstrate that our method leads to improvements across multiple open-source and closed-source models, validating the effectiveness of our approach.
>
> | Model | Acc.  |
> | ----- | ----- |
> | Llama 2-7b  | 46.23 |
> | **Llama 2-7b + Ours**  | **49.00 (2.77 ↑)** |
> | Llama 3-70b | 53.25 |
> | **Llama 3-70b + Ours** | **58.64 (5.39 ↑)** |
> | GLM 4  | 60.17 |
> | **GLM 4 + Ours** | **64.00 (3.83 ↑)** |
> | Gemini 1.5  | 54.23  |
> | **Gemini 1.5 + Ours**  | **59.84 (5.61 ↑)** |
> | Claude 3.5  | 70.38 |
> | **Claude 3.5 + Ours**  | **75.60 (5.22 ↑)** |
> | | |
>
> #### 2) Model Size Effect
>
> Additionally, we conducted experiments on LLMs of different sizes to examine the impact of model size on the results. The findings are as follows. The results indicate that model size has a significant impact on performance. **Larger models generally yield better results, demonstrating the positive effect of increased model size.**
>
> | Size | Acc.  |
> | ----- | ----- |
> | Llama 2-7b + ours  | 49.00 |
> | Llama 3-8b + ours  | 51.27 |
> | Llama 3-70b + ours | 58.64 |
> | | |
>
> #### 3) Non-LLM Methods
>
> We also compared our approach with non-LLM methods to highlight the advantages of using LLMs. **The comparison reveals a substantial difference in performance, with our method achieving improvements ranging from 11.7% to 30.5%.** The relevant results can be found in the corresponding papers.
>
> | Model | Acc.  |
> | ----- | -- |
> | Structured Attention[1] | 44.50 |
> | ReasTAP-Large[2]  | 58.70 |
> | TAPEX-large[3] | 59.10 |
> | OmniTab-Large[4] | 63.30 |
> | Ours | 75.00 |
> | | |
>
> [1] Wang B, Titov I, Lapata M. Learning Semantic Parsers from Denotations with Latent Structured Alignments and Abstract Programs[C]//Proceedings of the 2019 Conference on Empirical Methods in Natural Language Processing and the 9th International Joint Conference on Natural Language Processing (EMNLP-IJCNLP). 2019: 3774-3785.
>
> [2] Zhao Y, Nan L, Qi Z, et al. ReasTAP: Injecting Table Reasoning Skills During Pre-training via Synthetic Reasoning Examples[C]//Proceedings of the 2022 Conference on Empirical Methods in Natural Language Processing. 2022: 9006-9018.
>
> [3] Liu Q, Chen B, Guo J, et al. TAPEX: Table Pre-training via Learning a Neural SQL Executor[C]//International Conference on Learning Representations.
>
> [4] Jiang Z, Mao Y, He P, et al. OmniTab: Pretraining with Natural and Synthetic Data for Few-shot Table-based Question Answering[C]//Proceedings of the 2022 Conference of the North American Chapter of the Association for Computational Linguistics: Human Language Technologies. 2022: 932-942.

---

> ### Author Response · Authors · 2024-11-21
>
> ### 2. **DP and Agent Joint**
>
> 1）Some LLM methods **also use a majority voting mechanism** for the results, but our approach achieved better outcomes.
>
> | Method | Result |
> | ---- | --- |
> | ReAcTable [5] | 68.00  |
> | Ours | 75.00  |
> | | |
>
> [5] Yunjia Zhang, Jordan Henkel, Avrilia Floratou, Joyce Cahoon, Shaleen Deep, and Jignesh M Patel. Reactable: Enhancing react for table question answering. Proceedings of the VLDB Endowment, 17(8):1981–1994, 2024c.
>
> 2）Our method **follows the SOTA approach, which combines DP and Agent.** We introduced triples , building upon the SOTA framework.
>
> We compared the results of the standalone DP and Agent modes, as well as the joint mode, with the SOTA results. As shown in the table below, **while the performance in the DP mode is roughly the same, significant improvements were observed in the Agent and joint modes. This demonstrates the effectiveness of incorporating triples.**
>
> For the comparable results in the DP mode, we speculate that this is due to DP relying on text-based reasoning, which has limited understanding of the structured nature of tables. Agent, using Python-based methods, has a stronger understanding of table structure.
>
> | Method | Result  |
> | ---------- | ------- |
> | (Liu et al., 2024a)-DP [6] | **66.99** |
> | Ours-DP | 66.51(0.48↓ almost same) |
> | (Liu et al., 2024a)-Agent [6]    | 63.77   |
> | Ours-Agent  | **68.72（4.95↑）** |
> | (Liu et al., 2024a)-DP&Agent [6] | 73.65  |
> | Ours-DP&Agent  | **75.00（1.35↑）**  |
> | | |
>
> [6] Tianyang Liu, Fei Wang, and Muhao Chen. Rethinking tabular data understanding with large language models. In Proceedings of the 2024 Conference of the North American Chapter of the Association for Computational Linguistics: Human Language Technologies (Volume 1: Long Papers), pp. 450–482, 2024a.
>
> The above is our response. Regarding the error classification you mentioned, we are currently organizing it and will provide an updated version later. Your suggestions have been very helpful and we will add them into the final version. If you have any further questions, please feel free to ask, and we will continue to respond. Thanks!

---

> ### Author Response · Authors · 2024-11-25
>
> ## 3. Error Analysis ##
> Here, we randomly selected 100 test samples and categorized the errors for analysis.
>
> | Description Losses | 1939/40 | 1940/41 | 1941/42 | 1942/43 | 1943/44 | 1944/45 | Total |
> | -- | -- | -- | -- | -- | -- | -- | -- |
> | Direct War Losses | 360,000 | -| -| -| -|183,000 | 543,000 |
> | Murdered | 75,000  | 100,000 | 116,000 | 133,000 | 82,000  | -| 506,000 |
> | Deaths In Prisons & Camps | 69,000  | 210,000 | 220,000 | 266,000 | 381,000 | -| 1,146,000 |
> | Deaths Outside of Prisons & Camps | -| 42,000  | 71,000  | 142,000 | 218,000 | - | 473,000 |
> | Murdered in Eastern Regions | - | -| -| -| -| 100,000 | 100,000 |
> | Deaths other countries | -| -| -| -| -| -| 2,000 |
> | Total | 504,000 | 352,000 | 407,000 | 541,000 | 681,000 | 270,000 | 2,770,000 |
> | | | | | | | | |
>
> **answer formatting errors:**
>
> | question：how many people were murdered in 1940/41? |
> | -- |
> | triple：(\\$1, were murdered in, year), (\\$2, number of people, \\$3) |
> | decompose：1. who were murdered in 1940/41? 2. what's the number of people who were murdered in 1940/41? |
> | reasoning：The people who were murdered are described in row 1 of the table. They were murdered in various years, including 1940/41, as indicated in the table. The number of people murdered in 1940/41 is 100000. Final Answer: 100000 |
> | gold answer：100,000 |
> | error analysis :100000 and 100,000 format different |
> | |
>
> **Incorrect numerical comparison:**
> | question:what is the only years that total losses were below 350,000? |
> | -- |
> | triple: (\\$1, total losses, below 350,000), (\\$2, were the only, years) |
> | decompose: 1. what is the total losses in each year? 2. which years had total losses below 350,000? 3. what is the only years that total losses were below 350,000? |
> | verification: The total losses in each year are as follows:  1939/40: 504,000; 1940/41: 352,000; 1941/42: 407,000; 1942/43: 541,000; 1943/44: 681,000; 1944/45: 270,000. The years that had total losses below 350,000 is: 1940/41. Final Answer: 1940/41 |
> | gold answer: 1944/45 |
> | error analysis: Incorrectly assumed that 352,000 < 350,000 |
> | |
>
> **special row errors:**
>
> | question: what is the last description of losses on this chart |
> | -- |
> | triple: (\\$1, had, most cyclists), (\\$2, finish within, the top 10), (\\$3, is, country) |
> | decompose: 1. who finished within the top 10? 2. what's the country for each cyclist? 3. which country had the most cyclists finish within the top 10? |
> | reasoning: Thought: Extract the unique values from the "Description Losses" column. Select the last value to answer the second sub-question. Action: python_repl_ast Action Input: df['Description Losses'].unique() Observation:  ['Direct War Losses' 'Murdered' 'Deaths In Prisons & Camps' 'Deaths Outside of Prisons & Camps' 'Murdered in Eastern Regions' 'Deaths other countries' 'Total'] Thought: The unique values of the "Description Losses" column are ['Direct War Losses', 'Murdered', 'Deaths In Prisons & Camps', 'Deaths Outside of Prisons & Camps', 'Murdered in Eastern Regions', 'Deaths other countries', 'Total']. Select the last value from this list. Action: python_repl_ast Action Input: df['Description Losses'].unique()[-1] Observation: Total Final Answer: Total |
> | gold answer: Deaths other countries |
> | error analysis: Unable to identify special rows, such as total. |
> | |
>
> | Date | Competition | Location | Country | Event | Placing | Rider | Nationality |
> | -- | -- | -- | -- | --  | -- | -- | -- |
> | 31 October 2008  | 2008–09 World Cup | Manchester | United Kingdom | Keirin | 2 | Jason Kenny | GBR |
> | ...  | ... | ... | ... | ... | ... | ... |
> | 1 November 2009  | 2009–10 World Cup | Manchester | United Kingdom | Team sprint | 1 | Jamie Staff | GBR |
> | | | | | | | |
>
> **counting errors:**
>
> | question: what is the number of 1st place finishes across all events? |
> | -- |
> | triple: (\\$1, is the number of, 1st place finishes), (across all events, total, \\$2) |
> | decompose: 1. for each event, what is the placing? 2. what is the total number of 1st place finishes across all events? |
> | reasoning: In event 0, Jason Kenny placed 2nd. In event 1, Victoria Pendleton placed 1st. In event 2, Victoria Pendleton placed 1st. In event 3, Jason Kenny placed 1st. In event 4, Jason Kenny placed 1st. In event 5, Ross Edgar placed 2nd. In event 6, Jamie Staff placed 1st. In event 7, Victoria Pendleton placed 1st. In event 8, Ross Edgar placed 1st. In event 9, Chris Hoy placed 1st. In event 10, Jason Kenny placed 1st. In event 11, Jamie Staff placed 1st. In event 12, Victoria Pendleton placed 1st. In event 13, Victoria Pendleton placed 1st. In event 14, Chris Hoy placed 1st. In event 15, Chris Hoy placed 1st. In event 16, Victoria Pendleton placed 2nd. In event 17, Chris Hoy placed 1st. In event 18, Ross Edgar placed 1st. In event 19, Jamie Staff placed 1st. The total number of 1st place finishes across all events is 14. Final Answer: 14 |
> | gold answer: 17 |
> | error analysis: count error, total is 17. |
> | |

---

> > ### Author Response · Authors · 2024-11-28
> >
> > Dear Reviewer BvQy,
> >
> > We have updated all the rebuttal responses in the newly submitted PDF. Due to page limitations, they are included in the appendix. Thank you again for your review. your suggestions have been incredibly helpful for improving our paper. We sincerely invite you to participate in the rebuttal process. Thank you!

---

### Official Review · Reviewer_yMYg · 2024-11-04

**Soundness:** 3
**Presentation:** 2
**Contribution:** 2
**Rating:** 8
**Confidence:** 3

**Summary:**

The authors use Triples in different phases of TableQA. The generation of triples during Question decomposition phase helps in extracting key subquestions and conversion of answers during answer verification phase helps remove LLM bias in verification and enables better answer verification.

**Strengths:**

1. Incorporating triples in multiple stages of TableQA is a novel idea and as shown by the evidence in the paper, helps improve the overall QA performance.
2. The authors demonstrate the effectiveness of TIDE by performing ablation study which is critical in identifying the impact of this method.
3. This method works alongside DP and adds an additional step of generating triples, so this keeps all the benefits of DP as well.

**Weaknesses:**

1. Inorder to demonstrate the impact of TIDE, it could have been useful to see TIDE with other standard LLMs as well. The ablation study shows the impact of TIDE but is this method tailored to work with GPT 3.5 in any specific way?

**Questions:**

1. Given that this is an additional step along the pipeline of TableQA steps, what does this step do to the average response time and number of tokens used?
2. Does this method of generating triples work for mathematical questions such as average age/salary etc? Does this mean that the subquestions would require triples for every entry in the table?

---

> ### Author Response · Authors · 2024-11-21
>
> We greatly appreciate your review. Your suggestions have been extremely helpful in improving our paper, and we will incorporate additional details in the final version to enrich its content. Below are our three responses to your questions, which we hope will provide more clarity about our work.
>
> ### 1. Experiments with Other LLMs
>
> In our paper, **we only present the results from GPT-3.5 to maintain fairness and consistency with previous LLM-based studies on the two datasets**. Here, we provide the result on other LLMs to show the efficiency of our method.
>
> #### 1) Other Open-Source and Closed-Source Models
>
> The results demonstrate that our method improves performance across multiple open-source and closed-source models, confirming its effectiveness.
>
> | Model                  | Acc.               |
> | ---------------------- | ------------------ |
> | Llama 2-7b             | 46.23              |
> | **Llama 2-7b + Ours**  | **49.00 (2.77 ↑)** |
> | Llama 3-70b            | 53.25              |
> | **Llama 3-70b + Ours** | **58.64 (5.39 ↑)** |
> | GLM 4                  | 60.17              |
> | **GLM 4 + Ours**       | **64.00 (3.83 ↑)** |
> | Gemini 1.5             | 54.23              |
> | **Gemini 1.5 + Ours**  | **59.84 (5.61 ↑)** |
> | Claude 3.5             | 70.38              |
> | **Claude 3.5 + Ours**  | **75.60 (5.22 ↑)** |
> | | |
>
> #### 2) Model Size Effect
>
> We also evaluated LLMs of different sizes to analyze the impact of model size on performance. The results are as follows. The results indicate that model size has a significant impact on performance. **Larger models consistently achieve better outcomes, highlighting the benefits of increased model capacity.**
>
> | Size        | Acc.  |
> | ----------- | ----- |
> | Llama 2-7b + ours | 49.00 |
> | Llama 3-8b + ours  | 51.27 |
> | Llama 3-70b + ours | 58.64 |
> | | |
>
> #### 3) Non-LLM Methods
>
> We also compared our approach with non-LLM methods to highlight the advantages of using LLMs. **The results show a substantial difference, with our method achieving improvements ranging from 11.7% to 30.5%.** The corresponding results can be found in the referenced papers.
>
> | Model                    | Acc.  |
> | ------------------------ | ----- |
> | Structured Attention [1] | 44.50 |
> | ReasTAP-Large [2]        | 58.70 |
> | TAPEX-large [3]          | 59.10 |
> | OmniTab-Large [4]        | 63.30 |
> | Ours                     | 75.00 |
> | | |
>
> [1] Wang B, Titov I, Lapata M. Learning Semantic Parsers from Denotations with Latent Structured Alignments and Abstract Programs[C]//Proceedings of the 2019 Conference on Empirical Methods in Natural Language Processing and the 9th International Joint Conference on Natural Language Processing (EMNLP-IJCNLP). 2019: 3774-3785.
>
> [2] Zhao Y, Nan L, Qi Z, et al. ReasTAP: Injecting Table Reasoning Skills During Pre-training via Synthetic Reasoning Examples[C]//Proceedings of the 2022 Conference on Empirical Methods in Natural Language Processing. 2022: 9006-9018.
>
> [3] Liu Q, Chen B, Guo J, et al. TAPEX: Table Pre-training via Learning a Neural SQL Executor[C]//International Conference on Learning Representations.
>
> [4] Jiang Z, Mao Y, He P, et al. OmniTab: Pretraining with Natural and Synthetic Data for Few-shot Table-based Question Answering[C]//Proceedings of the 2022 Conference of the North American Chapter of the Association for Computational Linguistics: Human Language Technologies. 2022: 932-942.

---

> ### Author Response · Authors · 2024-11-21
>
> ### 2. **average response time and number of tokens used**
>
> Our method involves two LLM calls: the first generates question triples and sub-questions, and the second generates answers and performs verification. Examples of the prompts are provided in the appendix, and the complete prompts are available in the supplementary materials.
>
> We analyzed the average time and token usage for the two LLM calls. Besides, compared to other LLM-based methods, **our approach reduces the number of API calls while achieving better results.**
>
> | Operation     | time   | token                                                        |
> | ------------- | ------ | ------------------------------------------------------------ |
> | decomposition | 2.190s | prompt length 1159 + table length +question length           |
> | verification  | 3.211s | prompt length 970 + table length +question length + triple and sub-question length |
> | | | |
>
>
> | Methods            | Result    | Number of API calls                                          |
> | ------------------ | --------- | ------------------------------------------------------------ |
> | CHAIN-OF-TABLE [5] | 59.94     | （Next operation 1+ Argument 1+ Transform table 1）*Iter N = 3N |
> | **Ours**           | **75.00** | **decompose triple and sub-questions 1+ Answer and verification 1 = 2** |
> | | | |
>
> [5] Zilong Wang, Hao Zhang, Chun-Liang Li, Julian Martin Eisenschlos, Vincent Perot, Zifeng Wang, Lesly Miculicich, Yasuhisa Fujii, Jingbo Shang, Chen-Yu Lee, et al. Chain-of-table: Evolving tables in the reasoning chain for table understanding. In The 12th International Conference on Learning Representations, 2024.
>
> ### 3. **generating triples work for mathematical questions**
>
> **Our method does not impose limitations, as it structures questions and answers into triples, making it broadly applicable to table question-answering tasks.**
>
> The WTQ dataset **covers a variety of operations and question types, including *where*, *which*, *what*, *who*, *how many*, *average*, and *is*, among others. It includes the mathematical questions you are concerned about.**
>
> Additionally, our triples are generated specifically based on the questions and are designed to aid in understanding the table context.
>
> To illustrate this, we have provided actual results from the test dataset:
>
> |      | Contestant         |  Age | Height                | Home City                          | Rank                    |
> | ---: | :----------------- | ---: | :-------------------- | :--------------------------------- | :---------------------- |
> |    1 | Nataša Marić       |   16 | 175 cm (5 ft 9 in)    | Liefering (originally from Serbia) | Eliminated in Episode 3 |
> |  ... | ...                |  ... | ...                   | ...                                | ...                     |
> |   15 | Bianca Ebelsberger |   24 | 179 cm (5 ft 10.5 in) | Aurach am Hongar                   | Eliminated in Episode 9 |
> | | | | | | |
>
> | question        | in cycle 4 of austria's next top model, what is the average of all the contestants' ages? |
> | --------------- | ------------------------------------------------------------ |
> | question triple | (in cycle 4 of austria's next top model, \\$1), (for each contestant, what is their age),  (\\$1, is the average of, contestants' ages) |
> | sub-questions   | 1. what is in cycle 4 of austria's next top model?  2. for each contestant, what is their age? 3. what is the average of all the contestants' ages? |
> | Reasoning | in cycle 4 of austria's next top model, there are 16 contestants, match (in cycle 4 of austria’s next top model, \\$1), yes.  the ages of the contestants are 16, 16, 16, 16, 17, 17, 17, 18, 20, 20, 20, 21, 21, 21, 23, 24, match (for each contestant, what is their age), yes. the average of all the contestants' ages is (16+16+16+16+17+17+17+18+20+20+20+21+21+21+23+24) / 16 = 18.94, match (\\$1, is the average of, contestants' ages), yes. Final Answer: 18.94|
> | | |
>
> The above constitutes our complete response. Your suggestions have been very helpful for improving our paper, and we will incorporate them into the final version. If you have any further questions, please feel free to ask, and we will continue to respond. Thank you!

---

> ### Author Response · Authors · 2024-11-25
> **Rebuttal Invitation**
>
> Dear Reviewer yMYg,
>
> We have provided our responses, and your suggestions have been highly valuable for improving our paper. The content of the rebuttal will be incorporated into the final version. If you have any further questions, please feel free to ask. Your recognition is very important to us. We sincerely invite you to participate in the rebuttal process. Thank you!

---

> > ### Author Response · Authors · 2024-11-28
> >
> > Dear Reviewer yMYg,
> >
> > We have updated all the rebuttal responses in the newly submitted PDF. Due to page limitations, they are included in the appendix. Thank you again for your review; your suggestions have been incredibly helpful for improving our paper. We sincerely invite you to participate in the rebuttal process. Thank you!

---

> ### Comment · Reviewer_yMYg · 2024-12-03
>
> Thank you for the response. I am satisfied with the responses. I have updated my score accordingly.

---

### Meta-Review · Area_Chair_tJqB · 2024-12-21

**Metareview:**

(a) Scientific Claims and Findings:
The authors propose TIDE, a method that uses triples to structure question decomposition and verification in TableQA tasks. The decomposition phase generates triples to reduce semantic gaps in sub-questions, and the verification phase uses these triples for explicit answer validation. Experiments demonstrate improvements over existing baselines on datasets like WTQ and TabFact, with state-of-the-art (SOTA) results.

(b) Strengths:

- Novel incorporation of triples into TableQA pipelines.
- Comprehensive experiments, including ablation studies, demonstrating the effectiveness of TIDE.
- Applicability to both direct prompting (DP) and agent-based reasoning modes.
- Improved performance across multiple LLMs and significant gains over non-LLM methods.

(c) Weaknesses and Missing Aspects:

- Limited exploration of computational overhead and the impact of model size on performance in the main text.
- Lack of discussion on potential data contamination in pretrained LLMs affecting results.
- Some concerns about whether TIDE’s gains stem from triple integration or added reasoning layers.

(d) Decision Factors:
The strong empirical results, comprehensive rebuttal, and clarifications provided during the review period outweigh concerns about initial clarity and limited novelty. The inclusion of additional experiments and detailed analyses on other LLMs strengthens the case for acceptance.

**Additional Comments On Reviewer Discussion:**

Points Raised by Reviewers:

- Reviewer yMYg: Asked for experiments on diverse LLMs, computational cost, and robustness for different question types.
- Reviewer BvQy: Questioned clarity of ablation studies and the contribution of triples versus additional reasoning layers.
- Reviewer uGLH: Highlighted the need for formal explanations of modules and deeper analyses of error propagation, verification failures, and non-LLM baselines.

Authors' Rebuttal and Changes:

- Conducted experiments with multiple LLMs, including open-source and smaller models, confirming TIDE’s effectiveness.
- Provided details on computational overhead, iteration limits for verification, and error rates.
- Added formal descriptions of submodules, ablation study implementation, and triple generation.
- Addressed concerns about data contamination by showing substantial improvements even compared to direct LLM-based QA.

Final Decision Considerations:

- Reviewers yMYg and uGLH increased their scores after rebuttal, indicating satisfaction with responses.
- Remaining concerns (e.g., novelty of integrating triples) are mitigated by TIDE’s clear empirical contributions and thorough comparisons.

---

### Decision · Program_Chairs · 2025-01-22

Accept (Poster)